# Emergence of a proton exchange-based isomerization and lactonization mechanism in the plant coumarin synthase COSY

Colin Y. Kim [1,2], Andrew J. Mitchell[1], David W. Kastner [2,3], Claire E. Albright[1], Michael A. Gutierrez [1], Christopher M. Glinkerman[1], Heather J. Kulik [3] & Jing-Ke Weng [1,4] ✉

Plants contain rapidly evolving specialized enzymes that support the biosynthesis of functionally diverse natural products. In coumarin biosynthesis, a BAHD acyltransferase-family enzyme COSY was recently discovered to accelerate coumarin formation as the only known BAHD enzyme to catalyze an intramolecular acyl transfer reaction. Here we investigate the structural and mechanistic basis for COSY's coumarin synthase activity. Our structural analyses reveal an unconventional active-site configuration adapted to COSY's specialized activity. Through mutagenesis studies and deuterium exchange experiments, we identify a unique proton exchange mechanism at the α-carbon of the *o*-hydroxylated *trans*-hydroxycinnamoyl-CoA substrates during the catalytic cycle of COSY. Quantum mechanical cluster modeling and molecular dynamics further support this key mechanism for lowering the activation energy of the rate-limiting *trans*-to-*cis* isomerization step in coumarin production. This study unveils an unconventional catalytic mechanism mediated by a BAHD-family enzyme, and sheds light on COSY's evolutionary origin and its recruitment to coumarin biosynthesis in eudicots.

Coumarins are 1,2-benzopyrone natural products found in many plants that serve diverse biological functions, including iron acquisition from the soil, plant defense against herbivores and pathogens, and regulation of plant–microbiome interactions[1–3]. In addition to their roles in plants' fitness, coumarins exhibit various pharmacological properties, such as antiviral[4], antibacterial[5], and anticoagulant activities[6], and represent an important pharmacophore in small-molecule drug development. Understanding genetic and biochemical mechanisms underlying coumarin biosynthesis in plants lays the foundation for expanding the utility of coumarins in agricultural and pharmaceutical applications[1,2].

Coumarins are biosynthetically derived from the plant phenylpropanoid pathway[2]. The *ortho*-hydroxylation of hydroxycinnamoyl-CoAs, catalyzed by Fe(II)- and 2-oxoglutarate-dependent dioxygenase

feruloyl-CoA 6′-hydroxylase (F6′H), is a key step in coumarin biosynthesis, as it creates a branch point in phenylpropanoid metabolism, directing a portion of the hydroxycinnamoyl-CoA intermediates towards coumarin production[7]. The subsequent *trans*-to-*cis* isomerization and lactonization of the derived *o*-hydroxylated hydroxycinnamoyl-CoAs to form coumarins had been thought to occur spontaneously[8]. Recently, Vanholme et al. reported the identification of coumarin synthase (COSY) from *Arabidopsis thaliana*, which exhibits specific enzymatic activities that accelerate coumarin formation from *o*-hydroxylated *trans*-hydroxycinnamoyl-CoA substrates[8]. Consistent with COSY's in vitro biochemical activities, *A. thaliana cosy* mutants show reduced levels of scopoletin and sideretin, which are major coumarins in Arabidopsis, and various iron-deficiency growth phenotypes when grown in iron-deficient soil[8].

[1]Whitehead Institute for Biomedical Research, Cambridge, MA 02142, USA. [2]Department of Biological Engineering, Massachusetts Institute of Technology, Cambridge, MA 02139, USA. [3]Department of Chemical Engineering, Massachusetts Institute of Technology, Cambridge, MA 02139, USA. [4]Department of Biology, Massachusetts Institute of Technology, Cambridge, MA 02139, USA. ✉e-mail: wengj@wi.mit.edu

COSY belongs to the BAHD acyltransferase family, which is named after the first four discovered enzymes of this family: benzyl alcohol *O*-acetyltransferase (BEAT)[9], anthocyanin *O*-hydroxycinnamoyl transferase (AHCT)[10], *N*-hydroxycinnamoyl/benzoyltransferase (HCBT)[11], and deacetylvindoline 4-*O*-acetyltransferase (DAT)[12]. Canonical BAHD acyltransferases catalyze the transfer of the acyl group from an acyl-CoA thioester donor molecule to an -OH or -NH$_2$-containing acyl acceptor molecule[13]. All canonical BAHD acyltransferases known to date contain a ubiquitously conserved catalytic histidine. In the ester-forming BAHDs, the catalytic histidine serves as a general base to deprotonate the -OH of the acyl acceptor substrate, priming it for nucleophilic attack on the acyl-CoA substrate[14]. Many BAHD acyltransferases also contain a highly conserved tryptophan residue in the active site, with its indolic nitrogen serving as an oxyanion hole to stabilize the tetrahedral intermediate formed after the initial nucleophilic attack[14]. With its specific role in coumarin biosynthesis, COSY is the only BAHD acyltransferase that uses a single substrate as both the acyl donor and acceptor to produce a lactone product. In addition, COSY may also facilitate *trans*-to-*cis* isomerization of *o*-hydroxylated hydroxycinnamoyl-CoAs prior to lactonization as part of its catalytic cycle, implicating a potential reconfiguration of the ancestral canonical BAHD acyltransferase catalytic machinery to accommodate its unique coumarin synthase activity[8].

To understand the mechanistic basis for the unconventional activity of COSY as a BAHD-family enzyme, we present several crystal structures of *A. thaliana* COSY (*At*COSY) in its apo and various ligand-bound forms. Through comparative structural analyses, site-directed mutagenesis, deuterium labeling, quantum mechanical (QM) cluster modeling, and molecular dynamics, we uncover the structural features and catalytic mechanism underlying COSY's unique coumarin synthase activity. Through phylogenomic analyses, we also query the potential evolutionary origin of COSY as a recently recruited catalyst in plant coumarin biosynthesis.

## Results

### Structural features of *At*COSY

To investigate the molecular basis for the unique catalytic activity of COSY, we first obtained the apo structure of *At*COSY at 1.9 Å resolution (Supplementary Fig. 1). Similar to other BAHD acyltransferases[14], *At*COSY displays pseudo-symmetric N-terminal (residues 1–181) and C-terminal (residues 230–451) domains, connected by a linker loop (residues 182–229) (Supplementary Fig. 1). The *At*COSY active site is located at the interface between the two pseudo-symmetric domains, and features the conserved catalytic residues, His161 and Trp371, stemming from the N- and C-terminal domains, respectively (Supplementary Fig. 1). Additionally, we observed a single Ca$^{2+}$ ion that binds to the linker loop region and is coordinated by the main-chain carbonyl oxygens of Arg211 and Leu214 residues (Supplementary Fig. 1).

Next, we obtained product-bound states of *At*COSY with scopoletin and umbelliferone at 2.5 Å and 2.3 Å resolutions, respectively (Fig. 1a and Supplementary Figs. 2 and 3). Scopoletin and umbelliferone are common coumarins found in plants both containing a hydroxy substituent at C7, with scopoletin possessing an extra methoxy substituent at C6. Both compounds adopt a similar pose in *At*COSY's active site coordinated by a shared set of active-site-lining residues, notably Phe40, Tyr42, and Cys164 (Fig. 1a and Supplementary Figs. 2 and 3). The benzene ring of Phe40 forms a π−π stacking interaction with the *O*-containing heterocycle of scopoletin, while the side chains of Tyr42 and Cys164 form hydrogen bonds with the methoxy and hydroxy substituents of scopoletin, respectively (Supplementary Fig. 2). Multiple sequence alignment of three COSY orthologs together with three phylogenetically related hydroxycinnamoyl-CoA shikimate/quinate hydroxycinnamoyltransferase (HCT) orthologs indicates that Phe40 is shared between the two orthologous groups, whereas Tyr42 and Cys164 are uniquely conserved among COSY orthologs (Fig. 1b

and Supplementary Fig. 4). Nevertheless, close inspection of the *p*-coumaroyl-shikimate-bound *At*HCT structure (PDB ID: 5KJU)[14] revealed that Phe34 (corresponding to Phe40 in *At*COSY) is a surface-exposed residue not involved in substrate/product binding.

All previously published canonical BAHD acyltransferase structures harbor two substrate-binding pockets that accommodate binding of acyl donors and acyl acceptors, respectively[14–16]. However, when the scopoletin-bound *At*COSY structure is superpositioned with the *p*-coumaroylshikimate-bound *At*HCT structure (RMSD = 2.034 Å)[14], it is evident that the accessible acyl-acceptor-binding pocket of *At*HCT is occluded by an active-site loop (residues 392–400) in *At*COSY (Fig. 1b, c). *At*COSY's confined active-site configuration demonstrates the structural need for specialized coumarin synthase activity, which involves a single *o*-hydroxylated hydroxycinnamoyl-CoA substrate containing both the acyl donor and acceptor moieties (e.g., 6-hydroxyferuloyl-CoA; 6OHFCoA).

Despite multiple attempts of soaking and co-crystallization of *At*COSY with various substrate mimetics, including *p*-coumaroyl-CoA, feruloyl-CoA, dihydro-*p*-coumaroyl-CoA and cinnamoyl-CoA, we did not yield substrate-bound crystal structures. Instead, we were able to obtain the structure of *At*COSY in complex with free CoA at 2.3 Å resolution, which occupies the conventional BAHD family CoA binding site when superimposed with *p*-coumaroyl-CoA-bound *At*HCT (PDB: 5KJT) (Supplementary Fig. 5). We noticed that a single oxidation of Cys308 to sulfenic acid was present in the CoA-bound structure, suggesting that it may play a role as a redox-gate for the proper substrate entry (Supplementary Fig. 6a). Guided by both the CoA-bound and scopoletin-bound *At*COSY structures, we modeled the *trans*-6OHFCoA substrate in the COSY active site (Fig. 2d). This reconstructed pose of *trans*-6OHFCoA indicates that residues Phe40, Tyr42, His161, Cys164, and Trp371 engage close contacts with the substrate, and are likely involved in specific substrate binding and catalysis (Fig. 2d).

### Structure-guided biochemical characterization of *At*COSY

To enable functional characterization of COSY, we established a coupled assay system under light where the COSY substrate, 6OHFCoA or 2-hydroxy-*p*-coumaroyl CoA (2OHpCCoA), is first generated from its corresponding free acid, 6OHFA or 2-hydroxy-*p*-coumaric acid (2OHpCA), using *A. thaliana* 4-coumarate-CoA ligase 1 (*At*4CL1) in the presence of free CoA, ATP and Mg$^{2+}$. After 3 min of preincubation, *At*COSY is then added. The reaction proceeds for an additional 15 min before it is quenched by urea, followed by immediate analysis by LC-MS (Fig. 2a, Supplementary Figs. 7 and 8, and "Methods"). We observed substantial spontaneous formation of coumarin from 6OHFCoA or 2OHpCCoA in the absence of *At*COSY, while addition of *At*COSY resulted in significant acceleration of coumarin production (Fig. 2b, c and Supplementary Fig. 7), consistent with the previously reported observation[8]. Since we observed binding of a Ca$^{2+}$ ion in the *At*COSY structure, we tested whether the coumarin synthase activity may be dependent upon Ca$^{2+}$ or other divalent metals. Nevertheless, we observed no significant difference in coumarin synthase activity whether or not Ca$^{2+}$ is present in the assay buffer (Supplementary Fig. 9). Similarly, addition of Zn$^{2+}$, Ni$^{2+}$, Mn$^{2+}$, Co$^{2+}$, Fe$^{2+}$ or Cu$^{2+}$ to the assay buffer had little impact on the coumarin synthase activity (Supplementary Fig. 10), suggesting the observed calcium-binding site may be an artifact of crystallization. Moreover, based on our observation of Cys308 oxidation in the CoA-bound structure, we generated the C308A and C308S mutants, and assayed their activities in buffers containing a concentration gradient of hydrogen peroxide (Supplementary Figs. 6b and 8). Even so, varying redox environments had little effect on coumarin synthase activity for wild-type (WT), C308A, and C308S enzymes (Supplementary Fig. 6b). Thus, C308 oxidation is likely a crystallography artifact in our solved structures.

The ubiquitously conserved catalytic histidine is critical for BAHD acyltransferase activities[17]. Therefore, to investigate the role of this

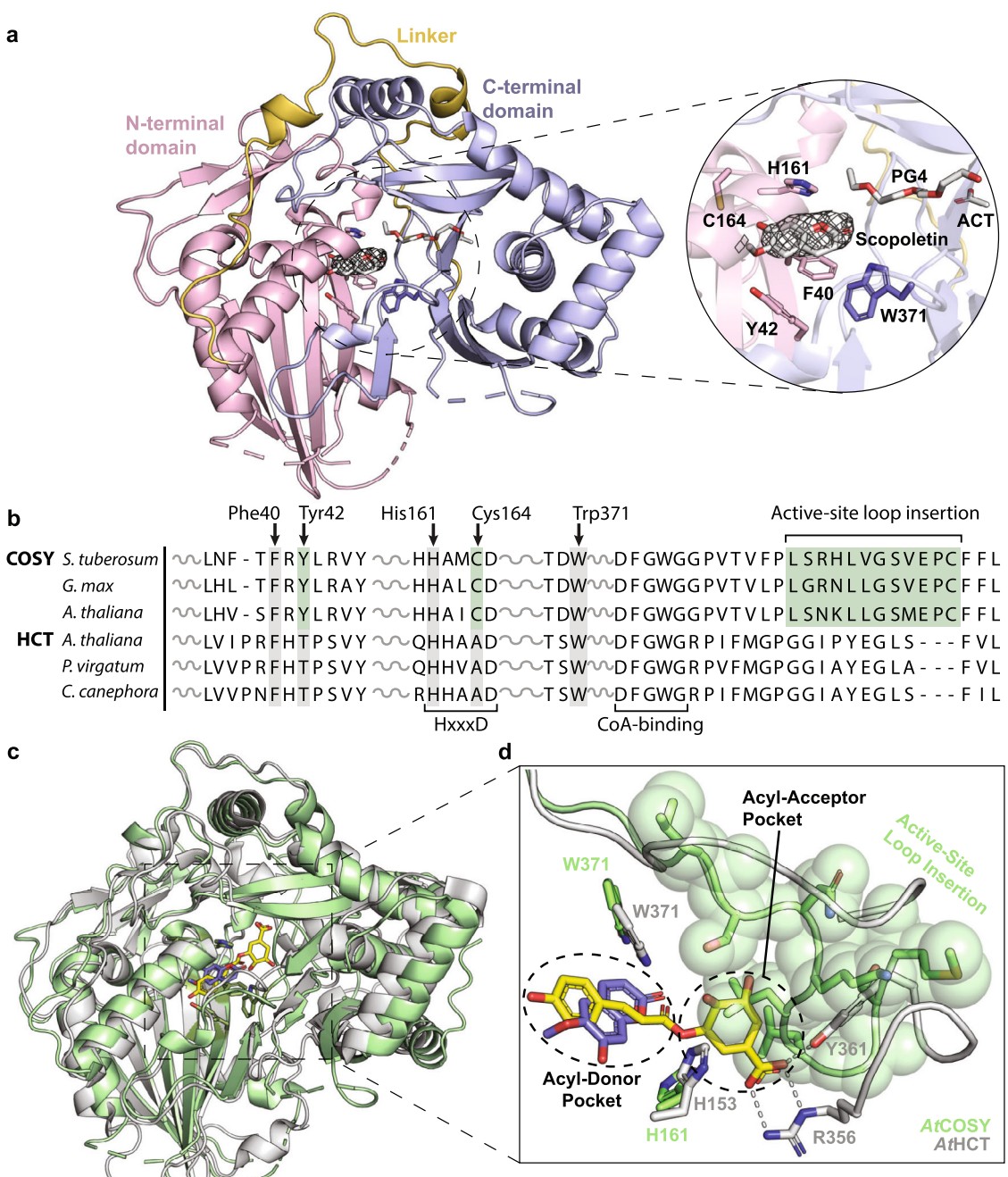

**Fig. 1 | Structural and sequence characterization of COSY in comparison with HCT. a** Scopoletin-bound *At*COSY at 2.5 Å resolution. The overall fold showcases a pseudo-symmetric N-terminal (residues 1–181; light pink) and C-terminal (residues 230–451; light blue) domains, connected by a linker loop (residues 182–229; yellow). The conserved BAHD catalytic residues (His161 and Trp371), along with product stabilizing residues (Phe40, Tyr42, and Cys164) are showcased within the active site pocket. Tetraethylene glycol (PG4) and acetate ion (ACT) are displayed. The mFo−DFc simulated annealing omit map with electron density map around SCO ligand is contoured at 1.0 σ. **b** Multiple sequence alignment of COSY orthologs from *Solanum tuberosum*, *Glycine max*, and *A. thaliana* with HCT orthologs from *A. thaliana*, *Panicum virgatum* and *Coffea canephora*. The full sequence alignment is shown in Supplementary Fig. 4. **c** Structural alignment of *At*COSY (green) and *At*HCT (PDB: 5KJU; gray). The product of *At*HCT, *p*-coumaroyl shikimate, is shown in yellow. Scopoletin is shown in purple. **d** Structural comparison between *At*COSY and *At*HCT reveals an active site loop insertion (residues 390–406) in *At*COSY that occludes the corresponding acyl acceptor pocket in *At*HCT.

residue in *At*COSY, we generated the H161A and H161Q mutants and assayed their activities using 6OHFCoA as the substrate. Surprisingly, the H161A and H161Q mutants exhibit WT-level coumarin synthase activity (Fig. 2e, Supplementary Figs. 8 and 11, and Supplementary Table 2). Although these results cannot rule out the possibility that His161 may still serve as a base that deprotonates the 6-OH group of 6OHFCoA as in other BAHD acyltransferases, His161 seems to be dispensable for the overall coumarin synthase activity, suggesting that

6-OH deprotonation is likely not a rate-limiting step in the full catalytic cycle of COSY.

Next, we examined the role of Trp371, which is the other highly conserved catalytic residue among many BAHD acyltransferases that serves as the oxyanion hole to stabilize the tetrahedral intermediate[14]. Four Trp371 mutants, W371H, W371A, W371V and W371M, were generated and assayed against 6OHFCoA as the substrate. Whereas the W371H mutant shows WT-level coumarin synthase activity, the W371A,

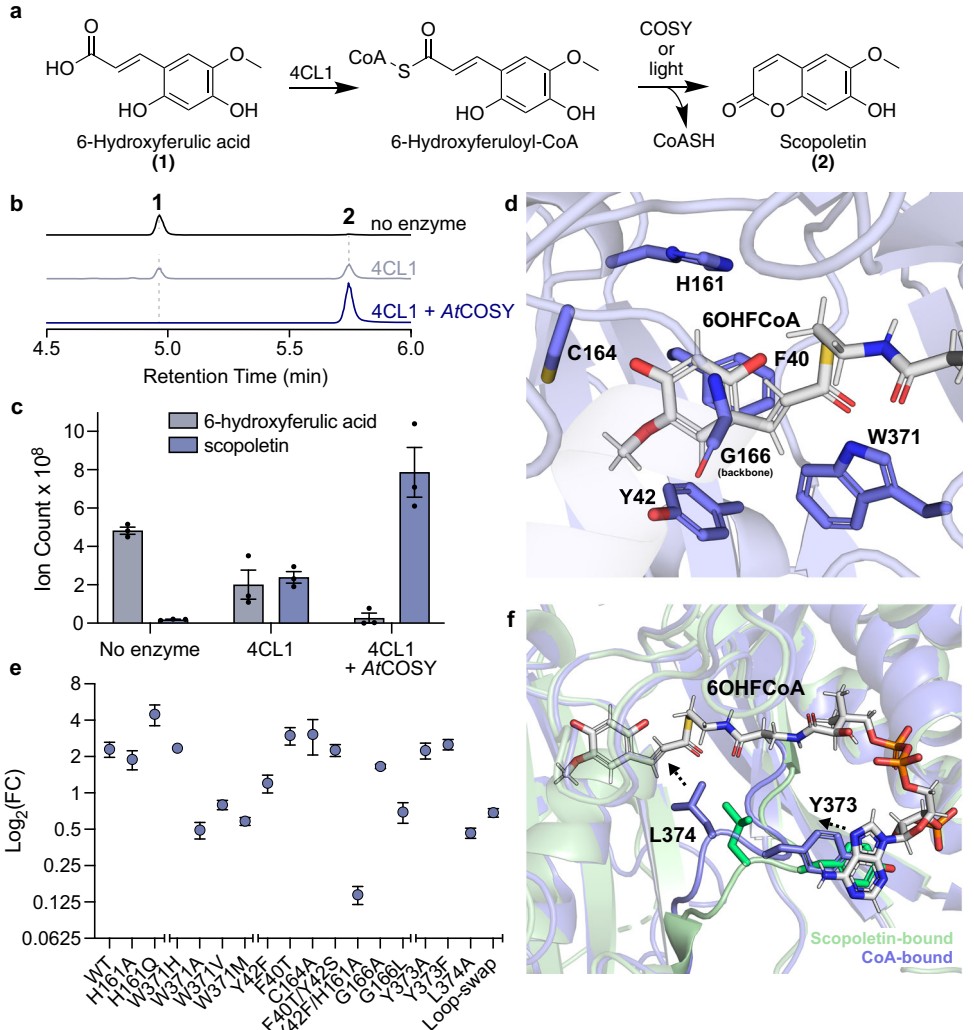

**Fig. 2 | In vitro coupled biochemical assay that examines the coumarin synthase activity of *At*COSY. a** Reaction schematic of the conversion of 6-hydroxyferulic acid (6OHFA) to scopoletin using 4-coumarate-CoA ligase 1 from *A. thaliana* (*At*4CL1) and coumarin synthase (*At*COSY). **b** Combined LC-MS extracted ion chromatograms (XICs) of 165.05556 *m/z* and 209.04549 *m/z*; **1**, and 191.03443 *m/z*; **2**. As scopoletin production is spontaneous, assay carried out in absence of *At*COSY serves as the baseline control for the coumarin synthase activity measurement. 4CL1 + *At*COSY sample shows complete conversion of 6OHFA into scopoletin. **c** Total ion count of 6OHFA and scopoletin from LC-MS for the enzymatic assays in triplicates. Data are presented as mean values +/− standard error of the mean. Relevant quantitative data underlying this figure is provided as a Source data file. **d** Reconstructed pose of *trans*−6OHFCoA in the active site of the CoA-bound *At*COSY structure. Phe40, Tyr42, His161, Cys 164, Trp371, and backbone atoms of Gly166 are highlighted as they make close contact with the substrate.

**e** Relative coumarin synthase activity of wild-type and mutant *At*COSY. Log₂ fold-change between the LC-HRAM-MS peak areas of coumarin ([M-H]⁻ = 191.03443 *m/z*) in 4CL1 + *At*COSY (WT or mutant) samples and that of 4CL1-only samples are shown. We performed the mutant assays for the following mutants (H161A, H161Q, W371H, W371A, W371V, W371M, Y42F, F40T, C164A, F40T/Y42S, Y42F/H161A, G166A, G166L, Y373A, Y373F, L374A, and "Loop-swap"). All assays were performed in triplicates and the error bars represent standard error of the mean (except for Y373A, which was performed in duplicate due to experimental error). Relevant quantitative data underlying this figure is provided as a Source data file. **f** Structural alignment of CoA-bound (blue) and scopoletin-bound (green) *At*COSY structures highlighting the dynamic loop containing Tyr373 and Leu374. Arrows indicate change in orientation of Tyr373 and Leu374 in CoA-bound structure compared to scopoletin-bound structure. Source data are provided as a Source Data file.

W371V and W371M mutants exhibit significantly reduced, but not completely abolished activities (Fig. 2e, Supplementary Figs. 8 and 11, and Supplementary Table 2). These results suggest that W371 plays an important but nonessential role in the catalytic cycle of COSY. W371 likely contributes to the catalytic cycle of COSY through its indole nitrogen, which can be substituted by the nitrogen-containing imidazole in the W371H mutant.

We then examined the roles of four additional active-site-lining residues: Phe40, Tyr42 and Cys164, which make close contact with the phenolic portion of COSY substrates and coumarin products, and Gly166, the backbone nitrogen of which is in hydrogen bonding distance to C2 of the 6OHFCoA substrate (Figs. 1a and 2d). We generated the F40T and C164A single mutants, as these substitutions are

observed between *At*COSY and *At*HCT structures at the corresponding positions (Supplementary Figs. 8 and 12). Additionally, we generated the Y42F mutant with the intention to abolish the hydrogen bond formed between the Tyr42 *p*-hydroxy and the methoxy group of scopoletin. To test the importance of the hydrogen bonding of Gly166 backbone nitrogen with C2, we also generated the G166A and G166L mutants (Supplementary Fig. 8). Furthermore, we constructed two double mutants F40T/Y42S and Y42F/H161A to perturb the COSY active site (Supplementary Fig. 8). The relative activities of these mutant enzymes were assayed against the 6OHFCoA substrate. Whereas the F40T, C164A, and F40T/Y42S mutants show WT-level coumarin synthase activities, the Y42F mutant shows a modest decrease in activity (Fig. 2e, Supplementary Fig. 11, and Supplementary

Table 2). These results suggest that none of the three residues are essential for the coumarin synthase activity of COSY despite their direct interactions with the substrate and product in the active site. The G166A mutant exhibits WT-level activity. In contrast, the steric bulk of leucine in the G166L mutant likely disfavors hydrogen bonding capability of its backbone nitrogen, leading to a significant decrease in coumarin synthase activity (Fig. 2e, Supplementary Fig. 11, and Supplementary Table 2). The Y42F/H161A double mutant did show a complete loss of coumarin synthase activity (Fig. 2e, Supplementary Fig. 11, and Supplementary Table 2), although this is likely due to overall impaired structural stability of this mutant. It was noted that the Y42F/H161A mutant was difficult to express and purify compared to WT AtCOSY and other mutants.

In search for other key catalytic residues of AtCOSY, we examined the various ligand-bound crystal structures for conformational changes of residues surrounding the active site. Indeed, we observed conformational differences of an active-site loop containing Tyr373 and Leu374 between the CoA-bound and scopoletin-bound AtCOSY structures. In the CoA-bound structure, Tyr373 exhibits π-π stacking interaction with the adenine portion of CoA causing Leu374 to shift towards C2 of 6OHFCoA, relative to its conformation in the scopoletin-bound structure (Fig. 2f and Supplementary Fig. 13). To test the role of this dynamic loop region, we generated the Y373A, Y373F, and L374A mutants and examined their activities against 6OHFCoA (Supplementary Fig. 8). The Y373A and Y373F mutants exhibit WT-level coumarin synthase activities, whereas the L374A mutant shows a significant decrease in activity similar to that of the W371A mutant (Fig. 2e, Supplementary Fig. 11, and Supplementary Table 2). These observations indicate that the π-π stacking interaction of Tyr373 with CoA does not contribute significantly to the coumarin synthase activity, while Leu374 on this dynamic loop region is likely involved in catalysis, albeit in a nonessential manner.

Altogether, our site-directed mutagenesis analyses of AtCOSY did not pinpoint specific residues that are absolutely required for the coumarin synthase activity. This is in stark contrast to conventional BAHD acyltransferases and other distantly related acyltransferases which require a critical basic residue to initiate the catalytic cycle as well as additional key catalytic residues for stabilizing the tetrahedral intermediates or leaving groups[14,18,19]. We next examined the role of the uniquely observed acyl-acceptor binding site occlusion in AtCOSY by constructing a mutant with its active-site loop occupying this space swapped with the corresponding region of AtHCT based on sequence and structural alignment (Fig. 1b, c and Supplementary Fig. 8). This 'Loop-swap' mutant exhibits a significant decrease in coumarin synthase activity when assayed against the 6OHFCoA substrate, while harboring no p-coumaroyl-shikimate-producing-activity when assayed for AtHCT activity (Fig. 2e, Supplementary Figs. 11 and 14, and Supplementary Table 2). With this observation and given that acyl acceptor deprotonation does not seem to constitute the primary energy barrier for coumarin formation from o-hydroxylated hydroxycinnamoyl-CoAs, we hypothesized that the main catalytic mechanism of COSY is to employ the overall geometry and dynamism of its active site to facilitate substrate trans-to-cis isomerization, and thus bring the reactive groups closer to trigger lactonization.

## Mechanistic basis for substrate trans-to-cis isomerization catalyzed by COSY

Vanholme et al. proposed a mechanism for the trans-to-cis isomerization of o-hydroxylated hydroxycinnamoyl-CoA substrates through intramolecular electron delocalization prior to lactonization[8]. However, we were intrigued by a previous study on scopoletin and scopolin biosynthesis in Manihot esculenta (cassava) showing that in planta feeding of E-cinnamic-2-d$_1$ acid resulted in a hydrogen/deuterium (H/D) exchange at C2 of the derived hydroxycoumarins[20]. This observation led us to hypothesize that enzyme-catalyzed coumarin

formation may involve acid-base catalysis at C2, which was not depicted by the mechanism proposed by Vanholme et al.[8].

To test this hypothesis, we synthesized D2-6OHFA with two deuteriums installed at C2 and C6, respectively, and used it as the hydroxycinnamic acid input in the At4CL1-AtCOSY coupled assay (Fig. 3a and Supplementary Figs. 15–17). In the control experiment without AtCOSY, spontaneous coumarin formation from D2-6OHFCoA yielded scopoletin showing the predominant ion of 193.04747 (m/z); **4**, suggesting the retention of both deuteriums (Fig. 3b, d and Supplementary Fig. 18). On the contrary, the COSY-catalyzed reaction yielded scopoletin showing the predominant ion of 192.04114 (m/z); **5**, suggesting the exchange of one deuterium with hydrogen at C2 (Fig. 3b, d). Presumably, the H/D exchange entails protonation of C2 followed by deprotonation or dedeuteronation. Interestingly, we noticed that 89.7% of the scopoletin product formed in the COSY-catalyzed reaction was **5**, which is much higher than 50%—the theoretical ratio if deprotonation or dedeuteronation occurs non-selectively, not considering kinetic isotope effect (KIE). When KIE is considered, since the zero-point energy of a C-H bond is higher than that of a C-D bond, deprotonation of the C-H bond at C2 would be favored if this occurred non-selectively. However, our data showed the opposite is true. This result suggests that the second-half deprotonation step (dedeuteronation in the case of deuterium-labeled precursor) occurs in a stereoselective manner in the COSY active site (Figs. 3d and 4). Additionally, this also indicates that the first-half protonation step of C2 occurs stereoselectively at the si face of the olefin, as non-selective protonation would result in a lower H/D exchange rate (Figs. 3d and 4).

To further test the C2 proton exchange mechanism mediated by COSY, we performed the reciprocal H/D exchange experiment by assaying AtCOSY against the 2OHpCCoA substrate in 66% D$_2$O. Consistent with the H/D exchange seen in the previous experiment using deuterium-labeled substrate, we observed significant exchange of C2 hydrogen with deuterium from D$_2$O in the resultant umbelliferone product (Supplementary Fig. 19). The presence of deuteron at C2 is further confirmed by NMR analysis of the purified umbelliferone product from the enzyme assay (Supplementary Fig. 20). In contrast, spontaneous umbelliferone formation from 2OHpCCoA in 66% D$_2$O in the absence of AtCOSY showed no H/D exchange (Supplementary Fig. 19).

To test whether COSY is responsible for the acid-base chemistry that occurs at C2 of coumarin biosynthesis in vivo, we grew WT A. thaliana (Col-0) and the cosy mutant (SALK_080878) on MS media containing 50% D$_2$O, and measured H/D exchange rate in the major coumarin product of Arabidopsis, scopoletin. Consistent with the results from the in vitro H/D exchange experiments, scopoletin produced in WT plants shows much higher deuterium labeling than that of the cosy mutant (Supplementary Fig. 21), supporting that AtCOSY is responsible for catalyzing the C2 acid-base chemistry in coumarin production in vivo.

We hypothesized that the COSY-dependent proton exchange phenomenon at C2 of coumarin biosynthesis is directly linked to COSY's catalytic function in mediating the trans-to-cis isomerization of the o-hydroxylated hydroxycinnamoyl-CoA substrate. To identify residues that may be involved in the acid-base catalysis at C2, we assayed various AtCOSY mutants against D2-6OHFCoA, and measured the relative H/D exchange rate in the resultant scopoletin product (Fig. 3c and Supplementary Fig. 18). Among the mutants tested, the W371A, G166L, and "Loop-swap" mutants show a major decrease of H/D exchange in scopoletin compared to that of the WT enzyme (Fig. 3c and Supplementary Fig. 18), which is correlated with their reduced overall COSY activity (Fig. 2e). Trp371 is positioned close to C2 of the reconstructed trans–6OHFCoA substrate in the active site of the CoA-bound AtCOSY structure (Fig. 2d), and therefore may play a role in proton exchange at C2 through a water molecule coordinated by its indole amine. The backbone nitrogen of Gly166 is close to C2 of

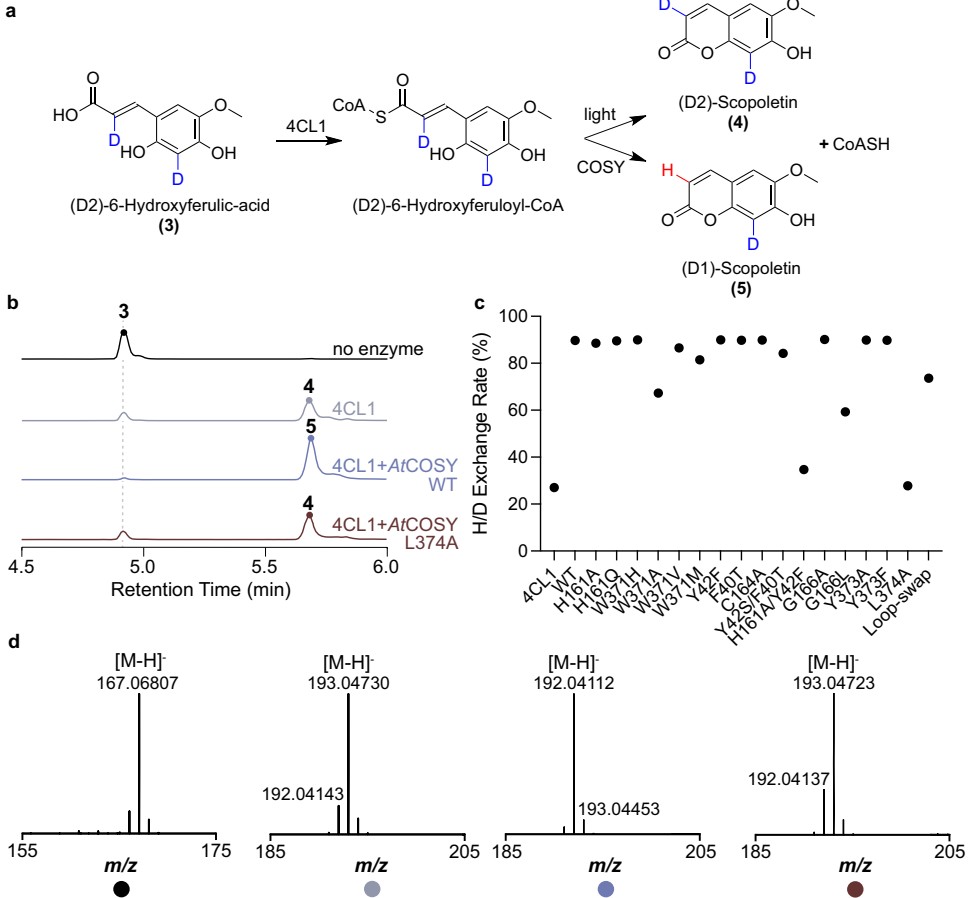

**Fig. 3 | In vitro coupled enzyme assay that examines the H/D exchange at C2 of the (D2)−6OHFA substrate. a** Reaction schematic for the conversion of **3**, (D2)−6-Hydroxyferulic-acid into **4**, (D2)-Scopoletin by light and **5**, (D1)-Scopoletin by COSY. **b** Combined LC-MS extracted ion chromatograms (XICs) of 167.06804 *m/z*; **3**, 193.04747 *m/z*; **4** and 192.04114 *m/z*; **5**. The colored circles represent the relevant substrate or product peaks in each XIC trace corresponding to the compound number in **a**. **c** H/D exchange rate of assays conducted with no *At*COSY, wild-type *At*COSY, and various *At*COSY mutants. H/D exchange rate is defined as the percentage of **5** out of the total ion intensity for *m/z* values corresponding to **4** and **5**. All assays were performed with 4CL1 in duplicates with data presented as mean values. Relevant quantitative data underlying H/D exchange is provided as a Source Data file. **d** MS[1] spectra of the relevant peaks from assays shown in **b** corresponding to compounds highlighted in **a**.

6OHFCoA and may engage in the hydrogen bonding network that stabilizes critical reaction intermediates in the catalytic cycle. The decrease in H/D-exchange rate observed in the 'Loop-swap' mutant is likely due to the overall alteration of the active-site geometry. Although the Y42F/H161A double mutant showed minimal H/D exchange barely above the no-*At*COSY control, we believe this is most likely due to its nearly abolished enzyme activity caused by structural instability (Fig. 3c and Supplementary Fig. 18). Among all mutants tested, the L374A mutant presents a unique case. Whereas the H/D exchange was completely abolished, the mutant enzyme still retains appreciable level of coumarin synthase activity, indicating a specific role of Leu374 in facilitating the proton exchange mechanism (Fig. 3b–d and Supplementary Fig. 18).

Based on new insights gained from the deuterium labeling experiments, we propose a revised catalytic mechanism for COSY. His161 acts as a nonessential base that promotes the deprotonation of the *o*-hydroxy of 6OHFCoA, leading to electron delocalization through the conjugated double bond system involving C2. The dynamic loop containing Leu374 helps position the substrate for the stereospecific protonation of C2 by a water molecule (Fig. 4a). The geometric orientation of this catalytic water molecule was confirmed by four replicates of 250 ns molecular dynamics (MD) simulations with the *At*COSY complex in the reactant state (Supplementary Fig. 22). The *s-trans* conformation of the substrate intermediate then undergoes rotary conformational change around the transient C2−C3 single bond

to arrive at the *s-cis* conformation. Next, Trp371 mediates a stereospecific deprotonation of C2 through a coordinated hydroxide, and in turn elicits lactonization and the formation of the tetrahedral intermediate with its oxyanion stabilized by the indole amine of Trp371. The catalytic cycle is completed by the release of CoA and the concomitant production of scopoletin. We reason that a key aspect of the catalytic function of COSY is to lower the energy barrier for the *trans*-to-*cis* isomerization of its substrate by enabling the alternative acid-base chemistry at C2 which does not happen in the uncatalyzed spontaneous reaction.

### Catalytic mechanism evaluation by density functional theory QM calculations

To further investigate the mechanistic role of key residues and the energetic feasibility of the proposed mechanism, we constructed large QM cluster models. Using the constructed QM models, we performed density functional theory (DFT) calculations on all proposed intermediates at the B3LYP/6-31 G* level of theory after confirming energetics were not sensitive to functional choice ("Methods" and Supplementary Table 3). The large QM cluster models include all residues in contact with the scopoletin and CoA portions of the substrate (Fig. 4b). The cluster model was constructed from the scopoletin and CoA-bound crystal structures (Supplementary Fig. 23). Before calculating the energetics, we identified five stepwise intermediates between the reactant, 6OHFCoA, and the products, scopoletin and

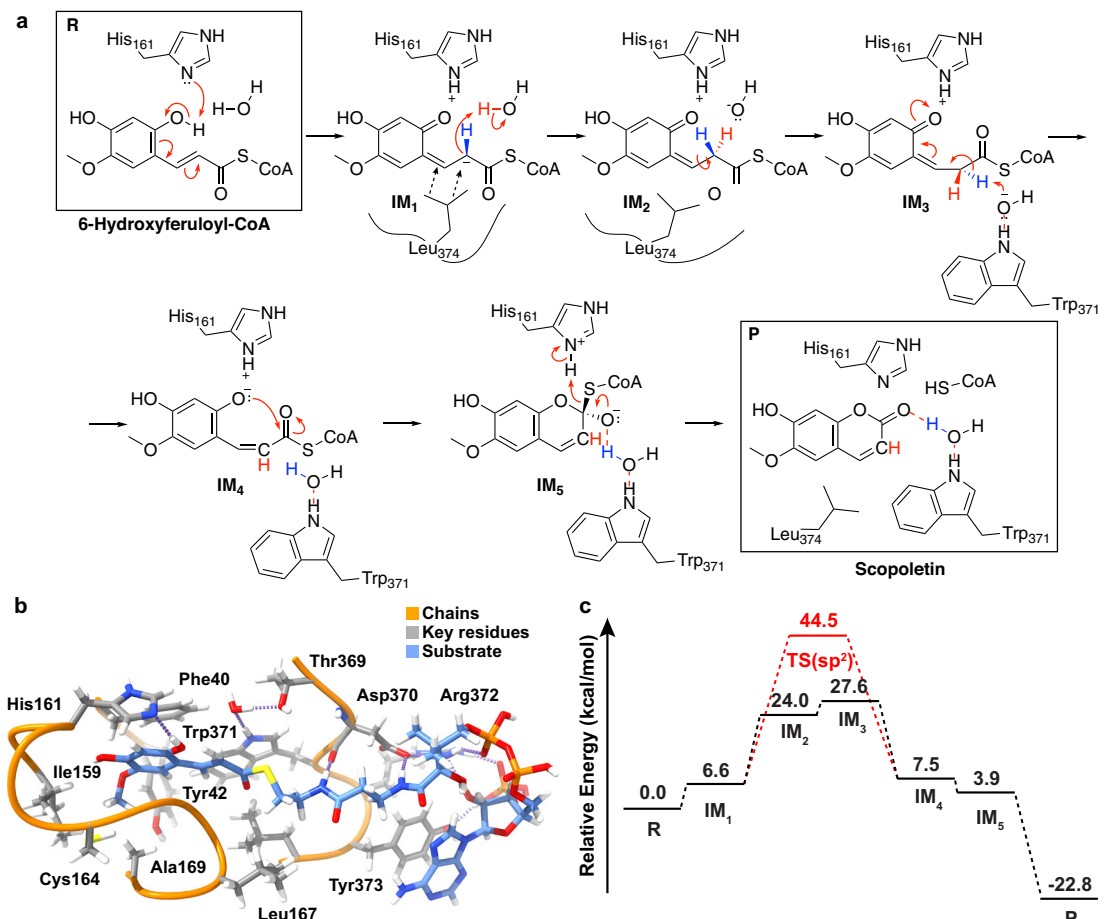

**Fig. 4 | Computational insights into the proposed intermediates and reaction energetics of COSY-catalyzed coumarin synthesis. a** A schematic of the proposed stepwise mechanism of COSY-catalyzed coumarin synthesis from 6OHFCoA. The reaction begins with deprotonation of the 6-OH of 6OHFCoA by His161, followed by delocalization of electrons through the conjugated double bond system involving C2 of 6OHFCoA. Then, Leu374 helps orient IM₁ towards the catalytic water molecule through van der Waals force, leading to the addition of a proton at C2. The C2-C3 bond then undergoes isomerization from the *s-trans* to the *s-cis* configuration. Following isomerization, lactonization and release of CoA lead to the generation of scopoletin. The oxyanion of IM₅ is depicted to be stabilized by a water molecule coordinated by Trp371, although the indole nitrogen of Trp371 may stabilize the oxyanion directly. Hydrogens that are relevant to stereoselective proton exchange are colored in red and blue. **b** The QM optimized large cluster model in the reactant state. Continuous chains of amino acids are shown as orange ribbons and included side chains are shown with gray carbons. The substrate is depicted with blue carbons. Hydrogen bonds are depicted as dotted purple lines. Relevant QM cluster model. xyz atomic coordinate file is provided as a Source data file. **c** The overall energetics for all intermediates of the proposed mechanism. Reactants are labeled as R and products are labeled as P. A putative transition state for the reaction mechanism in the absence of C2-protonation is illustrated in red.

CoA (Fig. 4a). Evaluation of the QM-optimized geometries for each intermediate revealed that the *s-cis* isomer is the highest-energy intermediate with an energy of 27.6 kcal/mol relative to the reactant state (R) (Supplementary Table 3). Conversely, the product has the lowest relative energy of −22.8 kcal/mol (Fig. 4c). We also observe that the intermediate following the initial deprotonation via His161 (IM₁) is energetically uphill by only 6.6 kcal/mol. The low thermodynamic cost is in line with our mutagenesis studies, which suggest that the formation of IM₁ is not rate-limiting. To evaluate whether there would be a kinetic barrier to the deprotonation of 6OHFCoA, we performed a constrained geometry scan of the deprotonation across the N⋯H reaction coordinate. We found that the dissociation of the hydrogen is effectively barrierless and not the rate-limiting step of the mechanism (Supplementary Fig. 24). These findings support our observed biochemical data that His161 is dispensable for the overall coumarin synthase activity.

We next sought to computationally evaluate the barrier height of the *s-trans*-to-*s-cis* (IM₂-to-IM₃) isomerization and the effect of having an sp² or sp³ carbon at C2. While our experimental results predict C2 protonation to be an essential preparative step to isomerization, a previously proposed mechanism suggests that isomerization may

occur with the conjugated substrate in IM₁ due to electron delocalization[8] (Supplementary Fig. 25a). Thus, we calculated barrier heights for the previously proposed IM₁-to-IM₄ isomerization and our proposed IM₃-to-IM₄ isomerization. The constrained geometry scans of the substrates in implicit solvent with dielectric constant of water show that isomerization without C2 protonation leads to a large energetic cost of 37.9 kcal/mol in addition to the 6.6 kcal/mol energy of IM₁ (Supplementary Figs. 25b, c). Conversely, C2 protonation results in a modest thermodynamic penalty of 6.3 kcal/mol with no additional activation barrier (Supplementary Figs. 25b, c). Moreover, in three replicates of 250 ns MD simulation of *At*COSY with the sp³-substrate starting at a dihedral angle of 180° for the C1−C2−C3−C4 dihedral, which isomerized to a dihedral of approximately 80° immediately during the equilibration phase, whereas the sp²-substrate failed to isomerize (Supplementary Fig. 26). Upon isomerization, the proposed IM₄-to-IM₅ lactonization can occur subsequently as showcased by the downhill energetics in the constrained geometry scan of the C-O bond in implicit solvent (Supplementary Fig 27). These results indicate that protonation leads to a more favorable transition and that the *s-trans*-to-*s-cis* isomerization is likely rate-limiting (Fig. 4c). We suspected that the favorability of the protonated system is due to lower double-bond

character than the conjugated species. To quantify the differences in the C2–C3 bond for both species, we evaluated the Mayer bond order, a QM measurement of bonding. The Mayer bond order analysis reveals significantly stronger bond strength for the conjugated species (1.39) than the C2-protonated species (0.77). The Mayer bond orders also correlate well with their respective bond lengths of 1.38 and 1.49 Å, suggesting significant double bond character that would disfavor isomerization of the conjugated species.

To better understand how the protein environment stabilizes the high-energy $IM_3$ intermediate, we performed hydrogen bond analysis by identifying bond critical points (BCP) from the quantum theory of atoms in molecules (QTAIM)[21,22]. The analysis reveals that a strong hydrogen bond between His161 and the substrate stabilizes the *s-cis* conformation by −20.1 kcal/mol. Additionally, a hydrogen bond that is only present in the *s-cis* isomer, between the backbone of Gly166 and the substrate, stabilizes the conformation by −8.4 kcal/mol (Supplementary Fig. 28). We also identified strong hydrogen bonds that stabilize the hydroxide species essential to the acid-base chemistry (Supplementary Fig. 28). The hydrogen bond between Thr369 and the hydroxide contributes −22.0 kcal/mol, and the hydrogen bond with Trp371 contributes −20.3 kcal/mol. The positioning of these strong hydrogen bonds allow for the necessary orientation of the hydroxide for both the $IM_1$-$IM_2$ and $IM_2$-$IM_3$ transitions (Supplementary Fig. 29). Although the *s-trans*-to-*s-cis* isomerization is the highest-energy transition, our results suggest that the COSY active site has evolved to stabilize the high-energy intermediate.

Based on our observations that the COSY active site is key for stabilizing the highest-energy intermediate, we sought to understand how the protein environment affects the favorability of the products relative to the reactant. Our calculations show that the total energetic contribution of all hydrogen bonds between the ligands and the protein environment is greater in the product state (−73.5 kcal/mol) than in the reactant state (−51.7 kcal/mol) (Supplementary Fig. 30). To further evaluate the role of COSY in favoring product formation, we sought to compute the energetic favorability of CoA release without the protein environment. Geometry optimization of the reactant and products, without the protein environment, reveals that product formation is only energetically favorable by −2.6 kcal/mol, further illustrating the importance of the protein environment in accelerating scopoletin formation (Supplementary Fig. 31). Moreover, we explored the dispensability of His161 as a catalytic base and calculated the reaction energetics for two alternative initial steps that utilize a solvent hydroxide as the base or an intramolecular hydrogen transfer of *o*-hydroxy hydrogen to C2 of 6OHFCoA. We found that the initial deprotonation step can be performed favorably by a solvent hydroxide (Supplementary Fig. 32). We also observe that the intramolecular deprotonation by the C2-C3 double bond is a possible pathway for His161-independent reaction (Supplementary Fig. 33).

Given the importance of acid-base chemistry in the proposed mechanism, we also calculated the reaction energetics for an alternative mechanism that avoids the simultaneous formation of the cationic histidine and an anionic hydroxide species. The alternative mechanism proceeds via a hydronium carrier and adds a step to the mechanism (Supplementary Fig. 34). However, we find that using hydronium as a carrier does not increase the favorability of the mechanism due to hydronium forming significantly weaker hydrogen bonds with Thr369 and Trp371 (Supplementary Fig. 35). Thus, the first proposed mechanism remains the most likely.

### Evolutionary origins of COSY and its biochemical role in question

The unique biochemical activity of COSY as a BAHD acyltransferase-fold protein and its role in coumarin biosynthesis led us to probe its evolutionary origin in plants. To do this, we first performed a maximum-likelihood phylogenetic analysis of *At*COSY together with its closely related homologs from other representative plants covering major lineages of land plants. This analysis revealed an apparent COSY clade, sister to a clade that includes the characterized *A. thaliana* spermidine disinapoyl acyltransferase (*At*SDT; AT2G23510) (Supplementary Fig. 36). The COSY clade contains homologs from both monocot and eudicot species (Fig. 5a). This is intriguing because coumarins have not been reported as widespread metabolites in monocots, whereas they are commonly found in eudicots[23]. To resolve this quandary, we performed phylogenetic analysis of feruloyl-CoA 6′-hydroxylase 1 from *A. thaliana* (*At*F6′H1), which catalyzes the first committed step of coumarin biosynthesis, together with its closely related homologs from other plants (Supplementary Fig. 37). In contrast to the phylogenetic pattern of COSY homologs, F6′H seems to have emerged in early eudicots and is absent in monocots (Fig. 5a).

To examine whether COSY homologs within the COSY clade contain coumarin synthase activity, we produced and biochemically characterized two COSY homologs from the eudicot species *Glycine max* (Soybean) and *Solanum tuberosum* (Potato), and one from the monocot species *Sorghum bicolor* (Sorghum), respectively (Supplementary Fig. 8). In vitro enzyme assays using recombinant proteins showed that *Gm*COSY, *St*COSY and *Sb*COSY contain coumarin synthase activity comparable to that of *At*COSY (Fig. 5b, c). Furthermore, we predicted the structures of *Gm*COSY, *St*COSY and *Sb*COSY using AlphaFold2.0[24], and compared their structural features to *At*COSY (Supplementary Fig. 38). We noticed a similar backbone fold and consistent positioning of most of the active-site-lining residues in these predicted structures compared to the solved apo-structure of *At*COSY. Altogether, these results suggest that COSY orthologs likely existed in the last common ancestor of monocots and eudicots, and predate the later emergence of coumarin biosynthetic pathway in eudicots, marked by the occurrence of F6′H. Such phylogenomic distribution pattern thus indicates that COSY may contain some additional ancestral biochemical activity prior to its recruitment to coumarin biosynthesis in eudicots. The biochemical and biological functions of COSY orthologs in monocots will be an interesting topic for future research.

## Discussion

COSY is an unusual BAHD-family enzyme. Unlike conventional BAHD acyltransferases, which catalyze transfer of the acyl group from an acyl donor molecule to an acyl acceptor molecule, COSY catalyzes an intramolecular acyl transfer reaction involving a single *o*-hydroxylated *trans*-hydroxycinnamoyl-CoA substrate containing both the acyl donor and acceptor moieties. In line with this unconventional activity, COSY has evolved a unique active-site geometry where the conventional acyl acceptor binding pocket of those typical BAHD acyltransferases is occluded by a unique active-site loop. Moreover, coumarin formation requires *s-trans*-to-*s-cis* isomerization of the hydroxycinnamoyl-CoA substrate to occur prior to subsequent lactonization. Although *trans-cis* isomerization of hydroxycinnamoyl-CoAs could occur non-enzymatically, we show that it indeed constitutes the most significant energy barrier in coumarin synthesis from the *o*-hydroxylated *trans*-hydroxycinnamoyl-CoA substrate, and that the COSY active site enables an exquisite proton exchange mechanism to lower the activation energy for this isomerization step. In contrast, deprotonation of the acyl-accepting -OH group, key to the catalytic mechanism of conventional BAHD acyltransferases, is no longer a major rate-limiting step in the full reaction cycle of coumarin formation. This notion is corroborated by the fact that the ubiquitously conserved catalytic histidine among BAHD acyltransferases can be mutated in COSY without a loss of its coumarin synthase activity. The updated catalytic mechanism of COSY described in this study (Fig. 4a) therefore improves upon Vanholme et al.'s proposal[8] and reveals previously unknown aspects of the coumarin formation reaction catalyzed by COSY.

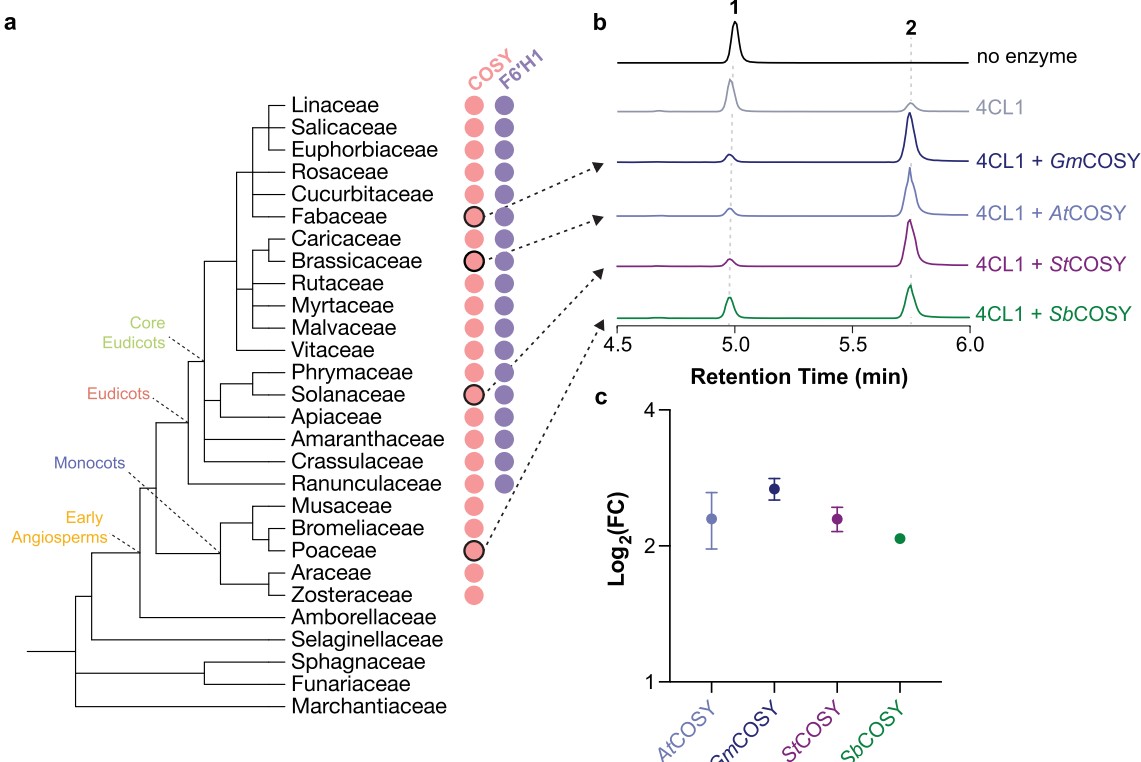

**Fig. 5 | Evolutionary origins of COSY in land plants. a** Presence of COSY (pink circles) and F6′H1 (purple circles) orthologs among major lineages of land plants. This information is inferred from maximum-likelihood phylogenetic analyses of COSY and F6′H1 together with their homologs from representative plants with sequenced genomes (Supplementary Figs. 28 and 29). **b** Combined LC-MS XICs of 165.05556 *m/z*; **1**, 209.04549 *m/z*; **1** and 191.03443 *m/z*; **2** showcasing the in vitro coumarin synthase activity of COSY orthologs from *G. max* (Soybean), *A. thaliana* (Arabidopsis), *S. tuberosum* (Potato), and *S. bicolor* (Sorghum). **c** Relative coumarin synthase activity of the four COSY orthologs quantified by the LC-HRAM-MS peak areas derived from the in vitro enzyme assay as shown in **b**. All assays were performed in triplicates. Data are presented as mean values with the error bars representing standard error of the mean.

Among the active-site-lining residues examined by site-directed mutagenesis, we identified Leu374, when mutated to alanine, completely abolishes the proton exchange mechanism (Fig. 3c), which is the main contributor to lowering the activation energy by the COSY active site. Interestingly, Leu374 is anchored on a dynamic loop that undergoes conformational change throughout the catalytic cycle. We postulate that the inward swing of Leu374 helps orient the reaction intermediate IM$_1$ in a catalytically favorable conformation for the subsequent protonation by water through van der Waals force (Fig. 4a). The important contribution of the dynamic Leu374 to the COSY catalytic mechanism is reminiscent of the previously described Arg356 residue in *At*HCT, where its positively charged side chain serves as a dynamic catalytic handle that engages an electrostatic interaction with the carboxyl group of the acyl acceptor substrate shikimate to help orient it towards the catalytic center for regio-selective acylation[14]. In both cases, these unique but critical structural and dynamic features found in these enzymes are highly specialized for the context of the reactions catalyzed by COSY and HCT, respectively, illustrating fine-grained mechanistic details underlying functional specialization within an enzyme family. Moreover, the observation that the L374A mutant still retains some coumarin synthase activity implicates additional contributions of the COSY active site to accelerate coumarin production besides the proton exchange mechanism. These contributions seem to be dispersed across numerous active-site-lining residues as revealed by our extensive site-directed mutagenesis analyses.

Beyond *trans*-to-*cis* isomerization of the hydroxycinnamoyl-CoA substrate in the context of COSY-catalyzed coumarin synthesis, *trans*-*cis* isomerization of olefin-containing compounds has been described in a wide range of biological processes. Examples include protein folding that involves the peptidyl-prolyl isomerase (PPIase)[25,26], bacterial adaptation to environmental stress through the *cis*-*trans* isomerase of unsaturated fatty acids[27], and regulation of cytokinin levels in plants that requires the activity of zeatin *cis*-*trans* isomerase[28,29]. Although *trans*-*cis* isomerization of olefins occurs spontaneously, sometimes at appreciable rates, precise regulation of these important biological processes relies on the flux balance of the involved isomeric compounds, which likely drove the evolution of discrete classes of isomerases that catalyze their production in a controlled manner. To the best of our knowledge, among the known catalytic mechanisms of various classes of olefin *trans*-*cis* isomerases, the acid-base proton exchange-based isomerization mechanism discovered in the context of COSY is unique, which exemplifies how discrete enzyme families could explore alternative catalytic mechanisms to accelerate a common type of chemical reaction.

The unconventional catalytic activity of COSY as a BAHD-family enzyme is reminiscent of the recently reported isochorismoyl-glutamate pyruvoyl-glutamate lyase (IPGL) in the Arabidopsis salicylic acid (SA) biosynthetic pathway, encoded by the *ENHANCED PSEUDOMONAS SUSCEPTIBILITY 1* (*EPS1*) gene[30]. Despite being a BAHD-family enzyme, EPS1 catalyzes the production of plant defense hormone SA from isochorismoyl-glutamate, a reaction that otherwise occurs spontaneously at a slower rate[31]. Similar to the case of COSY, the active site of EPS1 has evolved unique geometry and dynamism to accommodate the highly specialized IPGL activity[31]. Unlike COSY, EPS1 further harbors substitutions to the highly conserved catalytic histidine and tryptophan residues, suggesting its derived catalytic function has entirely deviated from the ancestral acyltransferase activity[31]. Both EPS1 and COSY illustrate the mechanistic plasticity of the BAHD protein fold, which permits reconfiguration of its ancestral active site to

access alternative catalytic functions through adaptive mutations. Reconstruction of such plausible mutational trajectories is an interesting topic for future research. It is worth noting that similar research has been previously pursued on the chalcone isomerase (CHI) fold proteins, which demonstrated the feasibility of arriving at the enantioselective CHI activity through laboratory evolution starting from a noncatalytic progenitor scaffold[32–34].

Several questions regarding the biochemical activities and biological functions of COSY have yet to be answered. We show that although the catalytic histidine and tryptophan remain unmutated among COSY orthologs, they are apparently nonessential to the coumarin synthase activity of COSY. Interestingly, our phylogenetic analyses suggest that the emergence of COSY predates the emergence of the coumarin biosynthetic pathway, which developed as an offshoot of the general phenylpropanoid metabolism only in eudicots. Could COSY still retain some ancestral activity outside the context of coumarin biosynthesis? What are the biological functions of COSY orthologs in monocots where coumarin biosynthesis is absent? Moreover, our study of COSY identified its enzymatic activity in converting *trans*-phenylpropanoids (in the form of CoA thioester) to their *cis* form. This observation raises the question regarding the biosynthetic origins of various *cis*-phenylpropanoids previously reported in plants[35,36]. For example, *cis*-cinnamic acid occurs in many plants and acts as an auxin transport inhibitor that promotes lateral root growth[37–39]. In addition, *cis-p*-coumaroylagmatine found in *Albizzia julibrissin* Durazz was reported to contain nyctinasty-inducing activities[35]. Although the occurrence of *cis*-phenylpropanoids has long been attributed to spontaneous *trans-cis* isomerization, perhaps enzymes like COSY are yet to be discovered that catalyze these physiologically important isomerization reactions.

## Methods

### Materials and general methods

Scopoletin (CAS No. 92-61-5, Product No. S2500), umbelliferone (CAS No. 93-35-6, Product No. H24003), 2OHpCA (CAS No. 614-86-8, Product No. 663158) were obtained from Sigma Aldrich. 6OHFA was synthesized by mixing scopoletin with 2.0 M NaOH solution at 23 °C followed by vigorous stirring at 80 °C for 1 h, acidification to pH 5 at 23 °C via dropwise addition of 1.0 M HCl, and extraction with EtOAc. Column chromatography was conducted to obtain purified 6OHFA using a Buchi Pure C-815 Flash Automated Chromatography System equipped with Buchi FlashPure EcoFlex $SiO_2$ prepacked columns (40–63 μm). All reagents used in biochemical assays were obtained from Sigma Aldrich. Deuterium oxide (CAS No. 7789-20-0, Product No. DLM-4-10X1ML) was obtained from Cambridge Isotopes Laboratories. All other reagents in this study were obtained from Thermo Fisher Scientific.

### Protein expression and purification

*At*COSY (AT1G28680) was amplified from root cDNA of *A. thaliana* and cloned into protein overexpression vector pHis8-4 containing an *N*-terminal 8xHis tag followed by a tobacco etch virus (TEV) cleavage site (Supplementary Table 4). The verified pHis8-4b::*AtCOSY* construct was transformed into *E. coli* strain BL21 (DE3) for recombinant protein expression. A 1 L of terrific broth (TB) medium containing 50 μg/mL kanamycin was inoculated with 30 mL of an overnight starter culture. The resulting culture was grown with shaking at 200 rpm at 37 °C to an $OD_{600}$ of 0.6–0.8. Then protein expression was induced with 0.5 mM isopropyl β-D-1-thiogalactopyranoside (IPTG) followed by cold shock of the medium and subsequent growth with shaking at 200 rpm (18 °C for 18 h).

Cultures were harvested by centrifugation and the resulting cell paste (~10 g/L) was resuspended in lysis buffer (100 mM HEPES pH 8.0, 200 mM NaCl, 20 mM imidazole, 10% (vol/vol) glycerol, 1 mM dithiothreitol) containing 1 mg/mL lysozyme and 1 mM

phenylmethylsulfonyl fluoride. Cells were lysed via five passes through an M-110L microfluidizer (Microfluidics) and the resulting crude protein lysate was clarified by centrifugation at $19,000 \times g$ for 1 h. Then the clarified lysate was loaded onto QIAGEN nickel-nitrilotriacetic acid (Ni-NTA) gravity flow chromatographic column. After loading the clarified lysate, the Ni-NTA resin was washed with 10 column volumes of lysis buffer, followed by 10 column volumes of lysis buffer with 5 mM ATP. Then the purified 8xHis-tagged proteins are eluted with 1 column volume of elution buffer (100 mM HEPES pH 8.0, 200 mM NaCl, 250 mM imidazole, 10% (vol/vol) glycerol, 1 mM dithiothreitol). To cleave the 8xHis-tag from purified protein, 1 mg of His-tagged TEV protease was added, followed by dialysis at 4 °C for 16 h in dialysis buffer (25 mM HEPES pH 8.0, 200 mM NaCl, 5% (vol/vol) glycerol, 0.5 mM EDTA, 0.5 mM dithiothreitol). After dialysis, the protein solution was passed through Ni-NTA resin to remove uncleaved protein and His-tagged TEV. The recombinant proteins were further purified by gel filtration on an ÄKTA Pure fast protein liquid chromatography (FPLC) system (GE Healthcare Life Sciences). The principal peaks were collected, verified by SDS polyacrylamide gel electrophoresis and dialyzed into a storage buffer (20 mM HEPES pH 7.5 and 5%(vol/vol) glycerol). Finally, proteins were concentrated to >10 mg/mL using Amicon Ultra-15 Centrifugal Filters (Millipore). For some of the mutants assayed in this study, 8xHis-tags were not cleaved, as this led to higher yield of purified proteins. Similar process was performed for the purification of *At*4CL1 (AT1G51680), and *At*HCT (AT5G48930). For construction and purification of COSY homologs *Gm*COSY, *St*COSY, and *Sb*COSY, the corresponding gene fragments were synthesized Integrated DNA Technologies (IDT) and followed a similar protocol (Supplementary Table 6).

### Crystallization and X-ray structure determination

Crystals for *At*COSY-apo, scopoletin-bound, umbelliferone-bound, and CoA-bound were grown at 25 °C by hanging-drop vapor diffusion method with the drop containing 2 μL of protein sample and 2 μL of reservoir solution at a reservoir solution volume of 500 μL. The crystallization buffer for all of the *At*COSY structures were composed of 50 mM calcium acetate, 0.1 M sodium cacodylate pH 6.5, and 20% glycerol. The protein sample concentration was 5 mg/mL stored in 20 mM HEPES pH 7.5. The ligand bound *At*COSY drop also contained 500 μM ligands. No additional cryo-protectant was added. The *At*COSY-apo structure was determined first by molecular replacement using the native HCT structure from *C. canephora* (PDB:4G0B) as the search model via the EMBL-HH Auto-Rickshaw[40,41]. The resulting model was iteratively refined using Refmac 5.2[42] and then manually refined in Coot 0.7.1[43]. For the subsequent *At*COSY structures, Phaser-MR[44] was used using the *At*COSY-apo structure as the mode and refined using RefMac 5.2[42] and Coot 0.7.1[43]. PHENIX 1.19.2 suite was used throughout for data processing and refinement, map generation, and preparing final models[45,46]. All structural models were further visualized using PyMOL 2.4.2. All crystallographic diffraction data sets were collected at beamlines 24-ID-C and 24-ID-E of the Advanced Photon Source at the Argonne National Laboratory by single-wavelength anomalous diffraction methods.

### Site-directed mutagenesis and in vitro biochemical assays

*At*COSY mutants were generated according to the protocol described in the QuickChange II Site-Directed Mutagenesis Kit (Agilent Technologies) using plasmid pHis8-4b::*AtCOSY* as the template and the primer sequences in Supplementary Table 5. pHis8-4b::*AtCOSY*-H161A was used as the template for generating Y42F/H161A double mutant. For the construction of active-site-loop insertion domain swap mutant, we replaced the DNA sequences encoding for [392]LSNKLLGSMEPC[403] in *At*COSY with that of the loop region in *At*HCT, [392]GGIPYEGLS[400]. The resulting gene fragment was synthesized and purchased from Twist Biosciences and was cloned into protein overexpression vector pHis8-

4 (Supplementary Table 6). The resulting mutant plasmid constructs were verified by sequencing. Recombinant mutant protein production and purification were carried out following the same procedure as described above.

Each in vitro biochemical assay for *At*COSY and its mutants with 6OHFA, D2-6OHFA, or 2OHpCA was carried out in 20 mM HEPES were performed with 1 mM hydroxycinnamic acid substrate, 1 mM ATP, 1 mM CoA, 0.5 mM CaCl₂ and 5 mM MgCl₂. 10 μM of *At*4CL1 was added to each assay 3 min before the addition of the respective enzyme at 2 μM. All assays were performed with a 15 min reaction period and subsequent 20 mM Urea quench. Each assay panel had an *At*4CL1-only and no enzyme control, along with three replicates of each target sample. Similar buffer conditions were used for the *At*HCT assay except 1 mM *p*-coumaric acid and 1 mM shikimic acid were used as substrates instead of hydroxycinnamic acid.

For enzyme assay in D₂O, the WT *At*COSY reaction was carried out following the same procedure as described above, except the assay was conducted in 66% D₂O. 66% D₂O was used as the stock solutions of the buffer components were already prepared in H₂O and 100% D₂O was not achievable for the in vitro assay. For the purification of umbelliferone-*d1*, we scaled-up this reaction with 2OHpCA. The resulting assay solution was concentrated to 1 mL and purified by preparative reverse-phase HPLC (Shimadzu Preparative HPLC with LC-20AP pump and SPD-20A UV–VIS detector) using Kinetex 5 μm C18 100 A, 150 × 21.2 mm column, solvent A-(H2O-0.1% TFA), solvent B-(CH3CN-0.1% TFA), 0–50 min 40–80% B, 10 mL/min) to provide umbelliferone-*d1* as a pale yellow oil. ¹H NMR spectra of umbelliferone and umbelliferone-*d1* were obtained using a Bruker Avance Neo 500 MHz spectrometer equipped with a 5 mm liquid nitrogen-cooled Prodigy BBO cryoprobe.

## Liquid chromatography and mass spectrometry

LC was conducted on a Vanquish Flex Binary UHPLC system (Thermo Fisher Scientific) using water with 0.1% formic acid as solvent A and acetonitrile with 0.1% formic acid as solvent B. Reverse phase separation of analytes was performed on a Kinetex C18 column, 150 × 3 mm², 2.6 μm particle size (Phenomenex). The column oven was held at 35 °C. Most injections were eluted with 5% B for 0.5 min, a gradient of 5–50% B for 6.5 min, 50–75% B for 0.5 min, 70% B for 0.5 min and 5% B for 2.0 min, with a flow rate of 0.5 mL/min. Most MS analyses were performed on a high-resolution Orbitrap Exploris 120 benchtop mass spectrometer (Thermo Fisher Scientific) operated in negative ionization mode with full scan range of 100–300 *m/z* and top four data-dependent MS/MS scans. The orbitrap resolution is 120,000 with RF lens of 70% and static spray voltage of 3500 V. Initial enzyme activity assays were analyzed with a TSQ Quantum Access Max mass spectrometer (Thermo Fisher Scientific) using multiple reaction monitoring at m/z transitions corresponding to [M-H]⁻ of 6OHFA (165.1 *m/z* to 150.1 *m/z*) and [M-H]⁻ of scopoletin (191.1 *m/z* to 176.1 *m/z)* with scan width of 0.5 and collision energy of 20 V. As 6OHFA and (D2)−6OHFA are readily fragmented under our MS collection methods, we monitored the abundance of their major fragment ions 165.05556 *m/z* and 167.06804 *m/z*, respectively, under negative ionization mode. For detection of D₂O labeling, single ion monitoring was used to scan the following *m/z* values: 161.05, 162.05, 165.03, 191.03, 192.04, 193.05, 209.05 with scan width of 0.5 and collision energy of 20 V. The MS analysis for quick detection of enzyme assay products was conducted on the TSQ Quantum Access Max mass spectrometer (Thermo Fisher Scientific) operated in negative ionization mode with a full scan range of 100–300 *m/z*. Raw LC-MS data were collected and analyzed using Chromeleon 7.2.10 ES, TSQ Tune 3.1.279.9, and XCalibur 4.5 (Thermo Fisher Scientific).

## Classical molecular dynamics simulations

The CoA-bound crystal structure (8DQR) was used for all MD simulations. The reactive state of the substrate was modeled using the conformation of scopoletin from the scopoletin-bound crystal structure (8DQP), which was then manually attached to CoA from 8DQR. Protonation states were added using the H++ webserver 4.0[47–49] with a pH of 7.0 and internal dielectric of 10 with all other defaults applied, while missing residues were added using Modeller[50]. The topology and coordinate files were prepared in AMBER 18 using tleap with the ff14SB force field[51]. The substrate was treated with general AMBER force field (GAFF) parameters and restrained electrostatic potential (RESP) charges[52] calculated with Gaussian 16.C.01 at the HF/6-31G* level of theory[53] with the RED RESP server[54]. All systems were solvated with 15 Å periodic TIP3P water boxes and neutralized with Na⁺ counterions[55]. The simulations were performed with the AMBER18 GPU-accelerated Particle Mesh Ewald molecular dynamics (PMEMD 18) code (pmemd.cuda)[56,57]. The Equilibration protocol is as follows: (i) hydrogen atom minimization (1000 steps), sidechain minimization with a fixed backbone (2000 steps), and unrestrained minimization (2000 steps). (ii) Controlled *NVT* heating was performed from 0 to 300 K for 10 ps using the Langevin thermostat and a collision frequency of 5.0 ps⁻¹, (iii) and 1 ns of *NpT* simulation using the Berendsen barostat and a 2 ps relaxation time. Following equilibration, 250 ns of production *NpT* dynamics were collected with SHAKE and 2 fs time steps.

## QM cluster model preparation

To investigate the energetic favorability of the proposed reaction mechanism, we performed geometry optimizations of large QM cluster models of the proposed intermediates[58]. While the crystal structures do not indicate the position of the reactant, we reconstructed 6OHFCoA by linking the substrates in the scopoletin-bound and the CoA-bound crystal structures after alignment with ChimeraX 1.3[59]. The scopoletin lactone ring was opened by changing the dihedral centered around the C2−C3 bond from *s-cis* to *s-trans*. The C1 carbonyl carbon was then connected to the thiol sulfur of CoA and the thiol hydrogen was removed. The manually adjusted atoms were then optimized with the MMFF94 force field with all other atoms fixed to conserve the crystallographic positions in preparation from QM cluster optimization. No 6OHFCoA atoms involved in substrate–protein contacts were modified during this process.

We next constructed a large QM cluster model that included all side-chains and backbone atoms in contact with the reactant substrate. The final cluster model included twenty amino acids, which corresponded to three residue chains: 40–42, 159–169, and 369–374. Full residues, including sidechain and backbone atoms, were Phe40, Tyr42, Ile159, His161, Cys164, Gly166, Leu167, Gly168, Ala169, Thr369, Asp370, Trp371, Arg372, Tyr373, and Leu374. Residues whose side chains did not interact with the substrate but whose backbone did, were substituted with glycine to minimize the size of the cluster model. Since backbone atoms were fixed to maintain the conformation of the active site, alanine and glycine substitutions are effectively the same. The substituted residues include Arg41, His160, Ala162, Ile163, and Asp165. A key water molecule proposed to be important in acid-base chemistry with the substrate, was modeled between Thr369 and Trp371. In total, the resulting cluster model contained 395 atoms. The substrate and protein environment both had charges of zero and maintained net charge neutrality. The starting QM cluster models are provided in the Source data. The backbone atoms for the terminal residues of each chain were capped with hydrogen atoms.

## QM cluster model calculations

QM cluster geometry optimizations and constrained geometry scans were performed with developer version 1.9 of the GPU-accelerated quantum chemistry package TeraChem[60–62]. All

calculations were carried out at the B3LYP/6-31G* level of theory. Additional calculations were carried out at the ωPBEh/6-31G* level of theory to verify that the results were insensitive to the functional (Supplementary Table 3). Given that the results were comparable for the range-separated hybrid functional, we will discuss the B3LYP results in the text. A moderate level of theory was selected to balance cost accuracy trade-offs and allow for a large QM cluster model that could account for all interactions between coumarin, CoA, and the protein environment. Given the large size of the cluster model, zero-point vibrational energy corrections were not computed and electronic energies are instead reported in the text. It is expected that this has a limited effect on conclusions from computational modeling because the computed differences between intermediates were large. To account for long-range interactions within the QM cluster model, semi-empirical DFT-D3 with default Becke-Johnson damping was also applied[62]. The calculations were carried out with implicit conductor-like polarizable continuum model (C-PCM) with epsilon set to 10 to approximate the electronic environment of the binding pocket[63]. To conserve the overall structure and crystallographic conformation of the active site during the optimization, backbone carbons were frozen. All other atoms were allowed to move freely. Putative transition states were identified using constrained geometry scans as implemented in TeraChem. During the scans, a selected reaction coordinate was fixed while optimizing all other degrees of freedom. In this implementation of the constrained geometry scan, each subsequent step uses the geometry of the previous step as input and the previous wavefunction as an initial guess to improve continuity. The strength of the hydrogen bonds present in each intermediate were estimated based on bond critical points (BCP)[64] from the quantum theory of atoms in molecules (QTAIM)[21] and the potential energy density calculated using the software package Multiwfn 3.8-dev[65]. To better understand the double bond character of the C2-C3 bond involved in the trans to cis isomerization, we performed Mayer bond order analysis using Multiwfn[22].

### Chemical synthesis and spectral verification of (*E*)-3-(2,4-dihydroxy-5-methoxyphenyl-3-*d*)acrylic-2-*d* acid (D2-6OHFA)

All reagents and solvents were used as supplied without further purification. Column chromatography was conducted using a Buchi Pure C-815 Flash Automated Chromatography System equipped with Buchi FlashPure EcoFlex SiO2 prepacked columns (40–63 μm). Analytical TLC was conducted using Millipore SiO2 60 F254 TLC (0.250 mm) plates. $^1$H and $^{13}$C NMR spectra were obtained using a Bruker Avance Neo 500 MHz spectrometer equipped with a 5 mm liquid nitrogen cooled Prodigy BBO cryoprobe. IR spectra were obtained using a Bruker Alpha 2 with a Platinum ATR accessory. Mass spectrometric analysis was performed on a ThermoFisher Scientific Orbitrap Exploris 120.

A vigorously stirred 2.0 M solution of NaOH in D2O (750 mL) at 23 °C was treated with scopoletin (25 mg, 0.13 mmol)[8] (Supplementary Fig. 12a). The reaction mixture was then warmed to 80 °C and stirred for 1 h. After 1 h, the reaction mixture was cooled to 23 °C, acidified to pH 5 by dropwise addition of 1.0 M HCl(aq), then extracted with EtOAc (3 × 1 mL). The combined organic extracts were dried over Na2SO4 and concentrated on a rotary evaporator. Flash chromatography (SiO2, 20–100% EtOAc/hexanes + 0.1% AcOH) provided **3** as a pale yellow oil (7.8 mg, 28%): $^1$H NMR (acetone-d6, 500 MHz) δ 7.98 (s, 1H), 7.19 (s, 1H), 3.85 (s, 3H); $^{13}$C NMR (acetone-d6, 125 MHz) δ 172.2, 168.9, 152.9, 151.1, 142.6, 141.0, 114.9, 114.8, 114.6, 114.4, 113.3, 111.8, 104.1, 104.0, 103.8, 103.6, 56.9; IR (film) $\nu_{max}$ 3267, 1661, 1582, 1495, 1455, 1433, 1268, 1199, 1121, 1061, 1038, 961, 666, 539, 515 cm$^{-1}$; HRMS (H-ESI-Orbitrap Exploris 120) *m/z* 211.05806 ($C_{10}H_8D_2O_5$–H$^+$ requires 211.05755).

### In planta deuterium exchange

*A. thaliana* wild type (Col-0) and *cosy* mutant (*AT1G28680*) seeds were obtained from Arabidopsis Biological Resource Center. All seeds were sterilized by washing with 70% (v/v) ethanol and 0.1% Tween 20, then with 100% ethanol. Seeds were subsequently washed with water to remove residual ethanol before vernalization at 4 °C for 2 days. The vernalized seeds were planted on Murashige and Skoog/agar media (half strength, 0.9% agar) prepared in 0% D2O or 50% D2O. 50% D2O was used because *A. thaliana* plants did not germinate or grow optimally on media prepared in 100% D2O, similar to previously reported[66]. Plants were grown under a 16-h-light/8-h-dark cycle with fluorescent light of 80 to 120 μmol photons m$^{-2}$ s$^{-1}$ at 22 °C and 60% RH. The roots of Col-0 and *cosy* plants were harvested after 20 days and flash frozen under liquid nitrogen. The frozen samples were homogenized on a TissueLyser II (QIAGEN) at 25 s$^{-1}$ for 2 min. Metabolites were extracted using 100 volumes (dry w/v) of 50% MeOH at room temperature for 15 min. Extracts were centrifuged once (13,000 × *g*, 10 min) to pellet plant tissue, and the supernatants were filtered using 0.2 μm filter vials (Thomson Instrument Company) for LC-HRAM-MS analysis.

### Phylogenetic analysis of COSY and F6'H

The amino acid sequences for phylogenetic analysis were obtained from curated plant proteomes in the Phytozome v12.1.6 database. NCBI pBLAST 2.12.0 searches with *At*COSY or *At*F6'H1 as query sequences were conducted and top 5 full length protein sequences were collected from each of the curated plant proteomes covering the angiosperm phylogeny and selected pre-angiosperm species. Multiple sequence alignments were built using the MUSCLE[67] algorithm in MEGA X[68]. Evolutionary histories were inferred using the maximum-likelihood method on the basis of the JTT matrix-based model in MEGA X[68]. Bootstrap statistics were calculated using 200 replicates. Similar procedure was followed for multiple sequence alignment of COSY and HCT homologous sequences, which was visualized using ESPript 3[69]. All alignment files can be found under Source data.

### Reporting summary

Further information on research design is available in the Nature Portfolio Reporting Summary linked to this article.

## Data availability

All atomic coordinates and structure factors generated in this study have been deposited in the Protein Data Bank database under accession codes 8DQO (COSY Apo), 8DQP (COSY+ scopoletin), 8DQQ (COSY+ umbelliferone), and 8DQR (COSY+ CoA). The native HCT structure from *C. canephora* used for molecular replacement was obtained from RCSB Protein Data Bank under identifier 4G0B. Relevant LC-HRAM-MS raw data files generated in this study have been deposited in Zenodo [https://doi.org/10.5281/zenodo.7513211]. The genomic sequences used for phylogenetic tree construction can be obtained in Phytozome v12.1.6 database and data underlying Supplementary Figs. 36 and 37 are provided in the Source data file. AlphaFold2.0 structural models generated for Supplementary Fig. 38 are also provided in the Source data file. Atomic coordinate files used in computational analyses are included in the Source data file. Data are also available from the corresponding author upon request. Source data are provided with this paper.

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

## Acknowledgements
We thank the staff at the CCP4/APS School for Macromolecular Crystallography for assistance with X-ray data collection and analysis. This work was supported by the W. M. Keck Foundation (J.-K.W.), the Family Larsson-Rosenquist Foundation (J.-K.W.), the Beckman Young Investigator Program (J.-K.W.), a Burroughs Wellcome Fund Career Award at the Scientific Interface (H.J.K., D.W.K), and a National Science Foundation Graduate Research Fellowship under Grant #1745302 (D.W.K.).

## Author contributions
C.Y.K., A.J.M., and J.-K.W. designed the research. C.Y.K., A.J.M., M.A.G., and J.-K.W. generated and processed the crystallographic data. C.Y.K., A.J.M., C.E.A., and M.A.G. performed the cloning, protein purification, and biochemical assays. C.Y.K. and C.E.A. conducted the *in planta* deuterium feeding study and phylogenetic analyses. C.M.G. performed chemical synthesis and characterization. D.W.K. and H.J.K. conducted and analyzed the QM cluster models and MD simulations. C.Y.K., D.W.K., H.J.K., and J.-K.W. interpreted the results and wrote the manuscript. All authors reviewed the manuscript.

## Competing interests
J.-K.W. is a member of the Scientific Advisory Board and a shareholder of DoubleRainbow Biosciences, Galixir and Inari Agriculture, which develop biotechnologies related to natural products, drug discovery, and agriculture. The remaining authors declare no other competing interests.
