## [Peer Review File · Nature Communications]

Emergence of a proton exchange-based isomerization and lactonization mechanism in the plant coumarin synthase COSYREVIEWER COMMENTS

Reviewer #1 (Remarks to the Author):

Kim et al. uncovered the proton exchange-based isomerization and lactonization mechanism for a known coumarin synthase COSY. The work reported apo and complex structures with the products for COSY, and studied the mechanism using deuterium labeling, site-directed mutagenesis, and quantum mechanical cluster modeling. The mechanism was unique compared to known trans-cis isomerases to date. Overall most part of the manuscript is clearly written. However, the work lacks of critical data to support the unique mechanism in its current form.

1. The crystal structure in a complex with the genuine substrate is preferred to explain the proton exchange-based mechanism. Results from docking are not always reliable to predict the substrate binding, especially when it is vital to the mechanism. This also applies to explaining the functions of the key amino acids in the mutants.
2. The mechanism proposed is not fully supported by the mutagenesis, as H161 was nonessential and W371 variants did not abolish the activities. Clearly the reaction relies on other key amino acids which has not been revealed.
3. While the conversion happens spontaneously and is catalyzed by light, it is not clear which of the enzyme activity assays were carried out in dark. It is thus difficult to understand certain results especially in the H/D exchange experiments.
4. The difference between COSY and conventional BAHD-family enzymes is attractive for the broad readership of Nature Communications. Conclusions of active-site loop and key amino acids need support from the reconstructed enzymes, and the diversity of the substrate is required.

Minor points

- 1) The structures of (D1)-scopoletin and (D2)-scopoletin should be confirmed by NMR.
- 2) The authors should explain why 66% D₂O and 50% D₂O were used in different H/D exchange experiments.

Reviewer #2 (Remarks to the Author):

In this article, the authors investigated the catalytic mechanism of a recently identified unique acyltransferase, COSY, through crystallographic analyses, site-directed mutagenesis, enzyme assay using deuterium labelled substrates, and computational chemistry. This class of acyltransferase is the only BAHD enzyme known to date that catalyzes intramolecular acyl group transfer. Therefore, explicating the catalytic machinery would substantially contribute to the elucidation of the potential roles of BAHD enzymes in plant secondary metabolism and could lead to unveiling undiscovered biosynthetic pathways.

However, there are a few questions I would like you to answer.

According to the proposed catalytic mechanism (Fig. 4), water (or hydroxide ion) seems to be one of the key factors. Since I cannot access to the structural data presented in this article now, it is impossible to confirm whether a water molecule was observed at the appropriate position in the crystal. If there is not, do you have any other experimental data that imply the presence of a water molecule at the position?

In the scheme, a water molecule is deprotonated to generate a hydroxide anion during IM₂–IM₃ transition. Then, one of the C₂ hydrogen is deprotonated stereo selectively by a hydroxide ion which is stabilized by W371. The latter hydroxide would be identical to the former hydroxide. But I wonder if the hydroxide ion generated from water is located at the distance and angle suitable for the following deprotonation from C₂ in the rotated form of the reactant (IM₃). It may be helpful if distance between the water molecule and C₂-hydrogen/carbon is indicated in the corresponding figures. Of course, it

would be better to prepare a new figure focusing on this point.

Is the following scenario feasible? — 6-Hydroxy group of 6OHFCoA is deprotonated by C2 and the hydrogen migrates to C2.

Additionally, I would like to see the energetic barrier to occurring spontaneous trans-cis isomerization and lactonization in water, if possible, at physiological pH for 6OHFCoA in plant.

Other comments:

1) Authors should address a reason why 6-OH-ferulic acid was monitored, in some experiments (e.g., Fig. 3, Fig. S10), by its fragment ion (e.g., m/z 165) instead of deprotonated molecule (e.g., m/z 209 $[M-H]^-$). In relation to this, it would be appreciated if the formula of ion ($[M-H]^-$, $[M+1-H]^-$, etc.) is given to each m/z value in the text to facilitate interpreting the methods and results. Additionally, the description “162.04623 m/z corresponding to the $[M-H+D]^-$ value of umbelliferone” in the legend to Fig.S17 would be incorrect. Since monoisotopic mass of umbelliferone is 162.0, m/z value of $[M-H+D]^-$ should be 163 (=162-1+2).

In page 28, “multiple reaction monitoring at m/z transitions corresponding to 6OHFA (166.1 m/z to 151.1 m/z) and scopoletin (192.1 m/z to 177.1 m/z) with...”: The m/z for the precursor ions would be 165 and 191 for 6OHFA and scopoletin, respectively, when analyzed by negative ionization mode.

2) In Fig. S23 and S24, it would be helpful if residue labels (one letter code and residue number) are shown for the representative residues, at least, those forming hydrogen bond with the ligand.

3) In legend to Fig. 3C, the definition of H/D exchange rate is somewhat confusing, because it is unclear what “total products” refers to. In the second and fourth panels of Fig. 3d, plural signals other than those originated from M+0, M+1, and M+2 appear to be present in some intensity. Are these ions also included in the products?

4) Please describe the definition of $\text{Log}_{10}(\text{FC})$ in Fig. 2e.

5) In Fig. 4a and Fig. S26, the drawing of curved arrows showing electron flow is insufficient or incorrect. The arrow at the phenoxide moiety of IM4 (Fig. 4a) should be deleted. In contrast, Fig. S26 depicts only some of the necessary arrows.

6) Accession No. of SbCOSY should be described. Although it may be appeared somewhere in the manuscript, I could not find it.

In Fig. S28, it would be better to label the branch tips corresponding to AtCOSY, GmCOSY, StCOSY, and SbCOSY.

7) In Abstract, “a unique proton exchange mechanism at the β -carbon” should be changed to “a unique proton exchange mechanism at the α -carbon”.

8) The term “hydroxyl” for the functional group -OH is not recommended by IUPAC. “Hydroxy” should be used instead.

Reviewer #3 (Remarks to the Author):

Kim and coworkers describe in their manuscript that COSY - contrary to previous hypotheses - possesses an unconventional active-site configuration.

The manuscript is well-written and the results are very interesting for the scientific community. The conclusions of the authors are sound and based on in-vitro enzyme activity test with deuterium-labeled substrates, enzyme mutagenesis studies and in silico quantum mechanical cluster modeling. The newly discovered mechanism has a substantial lower activation energy for the trans-to-cis isomerization of the hydroxycinnamoyl-CoA substrates, (a critical rate-limiting step leading to coumarin production) as compared to the mechanism previously proposed in Vanholme 2019. The results discussed in the manuscript are in line with earlier observations of scopoletin biosynthesis with

deuterated substrates as described by Bayoumi et al., 2008. An report that preceded the discovery of the COSY enzymes.

In addition, they made several interesting observations that are worth noticing. i) In vitro enzyme tests and mutagenesis studies showed that the observed Ca^{2+} ion and oxidation of Cys308 to sulfenic acid are not acceleration enzyme activity and are therefore likely artifacts. ii) The Phe40 seems to be involved in substrate/product binding in AtCOSY, but the Phe34 in AtHCT (corresponding to Phe40 in AtCOSY) is a surface-exposed residue and thus not most likely involved in substrate/product binding. iii) Mutation analysis showed that His161, although conserved and thought to be involved in 6-OH deprotonation, seems to be dispensable for the COSY activity, suggesting that deprotonation is not rate-limiting in the full catalytic cycle of COSY. This is further supported by QM calculations. iv) The authors correctly point out that the function of COSY orthologs in monocots will be an interesting topic for future research.

I have one major comment, that is likely easily address by the authors:

It is unclear to me whether X-ray diffraction was used to determine the structure of COSY, or whether the structure was computationally optimized based on the HCT. Although the authors mention the name "X-ray structure" in the materials and methods section and acknowledgments, the structure seems to generated via computational tools. "The AtCOSY-apo structure was determined first by molecular replacement using the native HCT structure from *C. canephora* (PDB:4G0B) as the search model via the EMBL-HH Auto-Rickshaw. [...]" Next there is also a section on the crystallisation; "Crystals for AtCOSY-apo, H161Q-apo, scopoletin-bound, umbelliferone-bound, and CoA-bound were grown at 25 °C by hanging-drop vapor diffusion method. [...]"

In the results section the authors write that crystallization was only successful for the CoA bound COSY: "Despite multiple attempts of soaking and co-crystallization of AtCOSY with various substrate mimetics, including p-coumaroyl-CoA, feruloyl-CoA, dihydro-p-coumaroyl-CoA and cinnamoyl-CoA, we did not yield substrate-bound crystal structures. Instead, we were able to obtain the structure of AtCOSY in complex with free CoA, [...]"

In case the CoA-bound structure of COSY was determined via X ray, then the information about the machines and techniques used need to be mentioned in the materials and methods section. The failed(?) attempt to generate crystals of other crystals could be useful to be mentioned. In case no X rays were used, it must be made clear from the results section that the structure of COSY was determined based on the structure of *C. canephora* HCT.

Minor comments:

- 1) A typo in the abstract: "courmarin production" \diamond "coumarin production"
- 2) 6OHFCoA is not defined
- 3) The concentrations of Zn^{2+} , Ni^{2+} , Mn^{2+} , Co^{2+} , Fe^{2+} or Cu^{2+} that are used for enzyme testing and their original chemical nature are not given.
- 4) I do not fully understand the colors used in the boxes of Figure 3b and d. Why do theses seem to correspond with the color of the peaks shown in the chromatograms and spectra? Please add the explanation in the legend of that figure.
- 5) "Evaluation of the QM-optimized geometries for each intermediate revealed that the s-cis isomer is the highest-energy intermediate with an energy of 27.6 kcal/mol relative to the reactant state (R) (Supplementary Table 3)." The name s-cis isomer is not used in the figures and tables, instead IM3 is used. Please define that s-cis is the same as IM3 at this position in the manuscript. (I noted later that it is defined in the next paragraph.)

REVIEWER COMMENTS

Reviewer #1 (Remarks to the Author):

Kim et al. uncovered the proton exchange-based isomerization and lactonization mechanism for a known coumarin synthase COSY. The work reported apo and complex structures with the products for COSY, and studied the mechanism using deuterium labeling, site-directed mutagenesis, and quantum mechanical cluster modeling. The mechanism was unique compared to known trans-cis isomerases to date. Overall most part of the manuscript is clearly written. However, the work lacks of critical data to support the unique mechanism in its current form.

1. The crystal structure in a complex with the genuine substrate is preferred to explain the proton exchange-based mechanism. Results from docking are not always reliable to predict the substrate binding, especially when it is vital to the mechanism. This also applies to explaining the functions of the key amino acids in the mutants.

We thank the reviewer for raising this point. We agree that a crystal structure of AtCOSY in complex with 6OHFCoA or 2OHpCA would present the pre-catalytic state preferred to explain the proton exchange-based mechanism, along with key amino acids in the mutants. However, as mentioned in our paper, we've attempted to soak and co-crystallize AtCOSY with the substrates, 6-hydroxyferuloyl-CoA, 2-hydroxy-*p*-coumaroyl-CoA, and various substrate mimetics, including *p*-coumaroyl-CoA, feruloyl-CoA, dihydro-*p*-coumaroyl-CoA and cinnamoyl-CoA. After extensive effort, we were not able to obtain substrate-bound crystal structures, which is likely due to the spontaneous reaction that poses challenges in capturing the substrate-bound state.

We recognize that our quantum mechanical (QM) cluster modeling results are not equal to crystal structures. However, they were modeled based on the precisely confined geometry of the crystal structures of AtCOSY-CoA (free CoA bound) and AtCOSY-scopoletin (product bound) structures. Thus, we argue that the 6OHFCoA-bound AtCOSY QM cluster models present a reasonable prediction of the substrate-bound state of the enzyme. In this revision, we have further used our QM cluster modeling results and our crystal structures to identify additional residues of AtCOSY, which may be involved in catalysis. The characterization of site-directed mutants at these residues are further discussed in response to the reviewer's comments #2 and #4.

We also apologize for the miswording in Figure 2d caption, we have corrected this caption to:

“(d) Reconstructed pose of *trans*-6OHFCoA in the active site of the CoA-bound AtCOSY structure...”

2. The mechanism proposed is not fully supported by the mutagenesis, as H161 was nonessential and W371 variants did not abolish the activities. Clearly the reaction relies on other key amino acids which has not been revealed.

We appreciate the reviewer for this concern. To computationally evaluate the experimental observation that His161 was non-essential, we ran an additional QM cluster model calculation with the H161A mutant, where hydroxide served as the catalytic base to deprotonate the 6-hydroxy group of 6OHFCoA. The calculations illustrated that the deprotonation step can be performed favorably by a solvent hydroxide. The reaction profiles and corresponding structures have been added to the manuscript as Supplementary Figure 32. We have added the following text to our manuscript:

“Moreover, we explored the dispensability of His161 as a catalytic base and calculated the reaction energetics for two alternative initial steps that utilize a solvent hydroxide as the base or an intramolecular hydrogen transfer of *o*-hydroxy hydrogen to C2 of 6OHFCoA. We found that the initial deprotonation step can be performed favorably by a solvent hydroxide (Supplementary Fig. 32).”

To further help clarify this concern raised by the reviewer, we sought to identify other potentially important residues for catalysis in AtCOSY. We recognized that the backbone nitrogen of Gly166 is in close proximity to C2 of 6OHFCoA and participates in specific hydrogen-bonding with the *cis*-isomer during the isomerization process (as shown in Supplementary Figure 28). Thus, we generated G166A and G166L AtCOSY mutants and tested their activities against 6OHFA and (D2)-6OHFA. While the G166A mutant did not exhibit any significant change in scopoletin production and H/D-exchange rate compared to WT AtCOSY, the G166L mutant portrayed a significant decrease in scopoletin production and H/D-exchange rate compared to WT. Nevertheless, we conclude that G166 is not an essential residue for catalysis, as we still observed enzyme-dependent production of scopoletin using G166L. These results are now incorporated in Figures 2, 3, Supplementary Figures 8, 11, 18, Supplementary Tables 2, 4, and 5 and we have added the following text to our manuscript:

“We then examined the role of four additional active-site-lining... , and Gly166, the backbone nitrogen of which is in hydrogen bonding distance to C2 of the 6OHFCoA substrate... To test the importance of the hydrogen bonding of Gly166 backbone nitrogen with C2, we also generated the G166A and G166L mutants (Supplementary Fig. 8). Furthermore, we constructed... The G166A mutant exhibits WT-level activity. In contrast, the steric bulk of leucine in the G166L mutant likely disfavors hydrogen bonding capability of its backbone nitrogen, leading to a significant decrease in coumarin synthase activity (Fig. 2e, Supplementary Fig. 11, Supplementary Table 2).”

“The backbone nitrogen of Gly166 is close to C2 of 6OHFCoA and may engage in the hydrogen bonding network that stabilizes critical reaction intermediates in the catalytic cycle.”

Additionally, we compared our obtained AtCOSY structures in apo, CoA-, umbelliferone-, and scopoletin-bound states, and observed a conformational change of an active-site loop containing Tyr373 upon binding CoA, where Tyr373 showcases π - π stacking interaction with the adenine portion of CoA. As a result of this conformational change, we noticed that Leu374 in CoA-bound structure is oriented towards the C2 of 6OHFCoA, potentially pushing on the substrate and favoring isomerization. Thus, we generated Y373A, Y373F, and L374A mutants and tested their activities against 6OHFA and (D2)-6OHFA. Y373A and Y373F did not show any significant change in scopoletin production and H/D-exchange rate compared to WT. L374A exhibited a significant decrease (but not complete abolishment) in scopoletin production compared to WT, and a complete abolishment of H/D-exchange compared to the 4CL1 sample that serves as a control sample for spontaneous formation of (D2)-Scopoletin. These results are incorporated in Figures 2, 3, Supplementary Figures 8, 11, 13, 18, Supplementary Tables 2, 4, and 5. Moreover, we decided to swap the Y42F/H161A mutant data in Figure 3b and 3d with the newly obtained L374A assay data, as it gave us a clearer depiction of the [M-D]⁻ *m/z* shift due to the AtCOSY-dependent H/D exchange at C2. We have added the following text to our manuscript:

“In search for other key catalytic residues of *At*COSY, we examined the various ligand-bound crystal structures for conformational changes of residues surrounding the active site. Indeed, we observed conformational differences of an active-site loop containing Tyr373 and Leu374 between the CoA-bound and scopoletin-bound *At*COSY structures. In the CoA-bound structure, Tyr373 exhibits π - π stacking interaction with the adenine portion of CoA causing Leu374 to shift towards C2 of 6OHFCoA, relative to its conformation in the scopoletin-bound structure (Fig. 2f, Supplementary Fig. 13). To test the role of this dynamic loop region, we generated the Y373A, Y373F, and L374A mutants and examined their activities against 6OHFCoA (Supplementary Fig. 8). The Y373A and Y373F mutants exhibit WT-level coumarin synthase activities, whereas the L374A mutant shows a significant decrease in activity similar to that of the W371A mutant (Fig. 2e, Supplementary Fig. 11, Supplementary Table 2). These observations indicate that the π - π stacking interaction of Tyr373 with CoA does not contribute significantly to the coumarin synthase activity, while Leu374 on this dynamic loop region is likely involved in catalysis, albeit in a nonessential manner.”

“Among all mutants tested, the L374A mutant presents a unique case. Whereas the H/D exchange was completely abolished, the mutant enzyme still retains appreciable level of coumarin synthase activity, indicating a specific role of Leu374 in facilitating the proton exchange mechanism (Figs 3b-d, Supplementary Fig. 18).”

“The dynamic loop containing Leu374 helps position the substrate for the stereospecific protonation of C2 by a water molecule (Fig. 4a).”

We identified Leu374 as a critical residue that is essential for the proton-exchange reaction catalyzed at C2, which is highly favorable for the reaction to proceed. Though we were unable to pinpoint a critical catalytic residue in *At*COSY that would completely abolish the activity (as seen in some other classical enzymatic systems), we continue to find that *At*COSY exemplifies an unusual case in that instead of relying on conserved key critical catalytic residues to mediate catalysis, its active-site behaves in a shape complementary, geometry-dependent approach to accelerate the *trans*-to-*cis* isomerization of 6OHFCoA. The fact that this reaction doesn't rely on key amino acids (i.e. the conserved catalytic histidine in BAHD-family enzymes) increases the structural novelty and mechanistic significance of COSY as a neofunctionalized BAHD-family enzyme. To this end, we have added the following text to our Discussion section:

“Among the active-site-lining residues examined by site-directed mutagenesis, we identified Leu374, when mutated to alanine, completely abolishes the proton exchange mechanism. Interestingly, Leu374 is anchored on a dynamic loop that undergoes conformational change throughout the catalytic cycle. We postulate that the inward swing of Leu374 helps orient the reaction intermediate IM_1 in a catalytically favorable conformation for the subsequent protonation by water through van der Waals force (Fig. 4a). The important contribution of the dynamic Leu374 to the COSY catalytic mechanism is reminiscent of the previously described Arg356 residue in *At*HCT, where its positively charged side chain serves as a dynamic catalytic handle that engages an electrostatic interaction with the carboxyl group of the acyl acceptor substrate shikimate to help orient it towards the catalytic center for regio-selective acylation¹⁴. In both cases, these unique but critical structural and dynamic features found in these enzymes are highly specialized for the context of the reactions catalyzed by COSY and HCT respectively, illustrating fine-grained mechanistic details underlying functional specialization within an enzyme family. Moreover, the observation that the L374A mutant still retains some coumarin synthase activity implicates additional contributions of the COSY active site to accelerate coumarin production besides the proton exchange mechanism. These contributions seem to be dispersed across numerous active-site-lining residues as revealed by our extensive site-directed mutagenesis analyses.”

3. While the conversion happens spontaneously and is catalyzed by light, it is not clear which of the enzyme activity assays were carried out in dark. It is thus difficult to understand certain results especially in the H/D exchange experiments.

This is a great point raised by the reviewer. Light catalyzed conversion of 6OHFA to scopoletin was reported in Vanholme et al. (Nat. Plants 5, 1066–1075, 2019) during 1 hr of incubation in light vs. dark. We conducted all our enzyme activity assays in light with the proper *At4CL1*-only control samples to compare and quantify the enzyme-specific conversion of 6OHFA to scopoletin. We also noticed that there is no significant difference when the reaction was carried out in light vs. dark conditions according to our assay protocol (3 min pre-incubation with *At4CL1* and 15 min incubation upon addition of *AtCOSY*-enzymes).

The following changes have been made in our manuscript to clarify this:

“To enable functional characterization of COSY, we established a coupled assay system under light where the COSY substrate, 6OHFCoA or 2-hydroxy-*p*-coumaroyl CoA ...”

In Figure 3a caption: “Reaction schematic for the conversion of **3**, (D2)-6-Hydroxyferulic-acid into **4**, (D2)-Scopoletin by light and **5**, (D1)-Scopoletin by COSY.”

4. The difference between COSY and conventional BAHD-family enzymes is attractive for the broad readership of Nature Communications. Conclusions of active-site loop and key amino acids need support from the reconstructed enzymes, and the diversity of the substrate is required.

We appreciate and agree with the reviewer’s comment that the difference between COSY and conventional BAHD acyltransferases is attractive for the broad readership of Nature Communications. To explore our conclusions more, we have identified additional amino acids relevant for catalysis and performed site-directed mutagenesis as written in response to the reviewer’s comment #2.

Furthermore, we generated a mutant harboring a domain-swap of the active-site loop present in *AtCOSY* (³⁹²LSNKLLGSMEPC⁴⁰³) with the corresponding loop region in *AtHCT* (³⁹²GGIPYEGLS⁴⁰⁰). This ‘Loop-swap’ mutant exhibited a significant decrease (but not complete abolishment of activity) in scopoletin production and H/D-exchange rate compared to WT.

The following text have been added in our main text to reflect this:

“We next examined the role of the uniquely observed acyl-acceptor binding site occlusion in *AtCOSY* by constructing a mutant with its active-site loop occupying this space swapped with the corresponding region of *AtHCT* based on sequence and structural alignment (Figs. 1b,c, Supplementary Fig. 8). This ‘Loop-swap’ mutant exhibits a significant decrease in coumarin synthase activity when assayed against the 6OHFCoA substrate, while harboring no *p*-coumaroyl-shikimate-producing-activity when assayed for *AtHCT* activity (Fig. 2e, Supplementary Figs. 11, 14, Supplementary Table 2). With this observation and...”

“The decrease in H/D-exchange rate observed in the ‘Loop-swap’ mutant is likely due to the overall alteration of the active-site geometry.”

Under Methods, Site-directed mutagenesis and *in vitro* biochemical assays: “For the construction of active-site-loop insertion domain swap mutant, we replaced the DNA sequences encoding for ³⁹²LSNKLLGSMEPC⁴⁰³ in *AtCOSY* with that of the loop region in *AtHCT*, ³⁹²GGIPYEGLS⁴⁰⁰. The resulting gene fragment was synthesized and purchased from Twist Biosciences and was cloned into protein overexpression vector pHis8-4 (Supplementary Table 6)... Similar buffer conditions were used for the *AtHCT* assay except 1 mM *p*-coumaric acid and 1 mM shikimic acid were used as substrates instead of hydroxycinnamic acid.”

To explore the diversity of the substrates, we tested the activity of *AtCOSY* on cinnamoyl-CoA and feruloyl-CoA, but did not observe any significant activity to report. We believe that this is beyond the

scope of the story presented in our paper and is tied to the future work surrounding the ancestral biochemical activity of COSY prior to its recruitment to coumarin biosynthesis.

Minor points

1) *The structures of (D1)-scopoletin and (D2)-scopoletin should be confirmed by NMR.*

We thank the reviewer for this suggestion. Due to the limited amount of D2-6OHFA substrate, it is challenging to obtain enough (D1)-scopoletin and (D2)-scopoletin in our enzyme assays for NMR. However, we believe that there is sufficient experimental evidence to support that the (D1)-scopoletin produced by AtCOSY-dependent activity has H/D-exchange occurring at C2. This is portrayed by our reciprocal H/D exchange experiment by assaying AtCOSY against 2OHpCCoA substrate in 66% D₂O as shown in Supplementary Figures 19 and 20, along with *in planta* H/D exchange experiment as shown in Supplementary Figure 21. Our observations also have been observed in Bayoumi et al. (*Phytochemistry* vol. 69 2928-2936, 2008) where feeding of *E*-cinnamic-2-d₁ to cassava roots resulted in a H/D exchange at C2 of the derived hydroxycoumarins. Altogether, we argue the presented data are sufficient to support that the [M-D] *m/z* value of 192.04112 observed in AtCOSY-dependent activity against (D2)-6OHFA is a result of the H/D-exchange at C2 as shown in Figure 3.

2) *The authors should explain why 66% D2O and 50% D2O were used in different H/D exchange experiments.*

We thank the reviewers for asking for clarification. We chose 66% D₂O in *in vitro* enzymatic assay conditions due to the inability to prepare buffers, substrates and other components of the assay in 100% D₂O. For *in planta* deuterium exchange experiments, we observed that *A. thaliana* plants are not able to grow optimally on ½ MS media prepared in 100% D₂O. We planted Col-0 and *cosy* seeds on a range of D₂O concentrations and found that 50% D₂O was the most optimal, highest concentration of D₂O.

The following changes have been made in our main text to reflect this:

Under Methods, Site-directed mutagenesis and *in vitro* biochemical assays:

“66% D₂O was used as the stock solutions of the buffer components were already prepared in H₂O and 100% D₂O was not achievable for the *in vitro* assay.”

Under Methods, *In planta* deuterium exchange:

“50% D₂O was used because *A. thaliana* plants did not grow on media prepared in 100% D₂O”

Reviewer #2 (Remarks to the Author):

In this article, the authors investigated the catalytic mechanism of a recently identified unique acyltransferase, COSY, through crystallographic analyses, site-directed mutagenesis, enzyme assay using deuterium labelled substrates, and computational chemistry. This class of acyltransferase is the only BAHD enzyme known to date that catalyzes intramolecular acyl group transfer. Therefore, explicating the catalytic machinery would substantially contribute to the elucidation of the potential roles of BAHD enzymes in plant secondary metabolism and could lead to unveiling undiscovered biosynthetic pathways.

However, there are a few questions I would like you to answer.

According to the proposed catalytic mechanism (Fig. 4), water (or hydroxide ion) seems to be one of the key factors. Since I cannot access to the structural data presented in this article now, it is impossible to confirm whether a water molecule was observed at the appropriate position in the crystal. If there is not, do you have any other experimental data that imply the presence of a water molecule at the position?

We thank the reviewer for this point. The water molecules captured in the active-site of our AtCOSY crystal structures are not in the proposed catalytic position. Though our crystallographic structures do not exactly capture the water in our proposed position, it doesn't negate the fact that the water could occupy that position. Water molecules are free flowing and not rigidly locked in place. Moreover, the same water molecule is not often involved in every step of the reaction mechanism. Water acid-base chemistry is very complex and there can be multiple water molecules involved. Our proposed single water molecule for the calculations could represent a possible catalytic process involving water.

Active site crystallographic waters in our structures indicate that water can enter the active site. To computationally investigate whether a water molecule could occupy the proposed site, we ran molecular dynamics simulations to observe the diffusion of water molecules through the active site. We ran four replicate 250 ns molecular dynamics simulations with the AtCOSY complex in the reactant state. In all replicates, we observed that the proposed site of the catalytic hydroxide was occupied and was solvent accessible. A representative snapshot was obtained by clustering the molecular dynamics simulations based on the conformation of the substrate, His161, Trp371, and Thr369. A representative snapshot of the active site in the predominant cluster has been included as Supplementary Figure 23 along with the following texts:

"The geometric orientation of this catalytic water molecule was confirmed by four replicate 250 ns molecular dynamics simulations with the AtCOSY complex in the reactant state (Supplementary Fig. 23)."

The details for molecular dynamics simulations are included under the Methods section:

Classical Molecular Dynamics Simulations

The CoA-bound crystal structure (8DQR) was used for all MD simulations. The reactive state of the substrate was modeled using the conformation of scopoletin from the scopoletin-bound crystal structure (8DQP), which was then manually attached to CoA from 8DQR. Protonation states were added using the H++ webserver⁴⁵⁻⁴⁷ with a pH of 7.0 and internal dielectric of 10 with all other defaults applied, while missing residues were added using Modeller⁴⁸. The topology and coordinate files were prepared in AMBER using tleap with the ff14SB force field⁴⁹. The substrate was treated with general AMBER force field (GAFF) parameters and restrained electrostatic potential (RESP) charges⁵⁰ calculated with Gaussian16 at the HF/6-31G* level of theory⁵¹ with the RED RESP server⁵². All systems were solvated with 15 Å periodic TIP3P water boxes and neutralized with Na⁺ counterions⁵³. The simulations were performed with the AMBER18 GPU-accelerated Particle Mesh Ewald molecular dynamics (PMEMD) code (pmemd.cuda)^{54,55}. The Equilibration protocol is as follows: i) hydrogen atom minimization (1000 steps), sidechain minimization with a fixed backbone (2000 steps), and unrestrained minimization (2000 steps). ii) Controlled NVT heating was performed from 0 to 300 K for 10 ps using the Langevin thermostat and a collision frequency of 5.0 ps⁻¹, iii) and 1 ns of NpT simulation using the Berendsen barostat

and a 2ps relaxation time. Following equilibration, 250 ns of production NpT dynamics were collected with SHAKE and 2 fs time steps.”

In the scheme, a water molecule is deprotonated to generate a hydroxide anion during IM₂–IM₃ transition. Then, one of the C2 hydrogen is deprotonated stereo selectively by a hydroxide ion which is stabilized by W371. The latter hydroxide would be identical to the former hydroxide. But I wonder if the hydroxide ion generated from water is located at the distance and angle suitable for the following deprotonation from C2 in the rotated form of the reactant (IM₃). It may be helpful if distance between the water molecule and C2-hydrogen/carbon is indicated in the corresponding figures. Of course, it would be better to prepare a new figure focusing on this point.

We thank the reviewer for this suggestion. To illustrate that the hydroxide involved in steps IM₁–IM₂ and IM₃–IM₄ can be the same hydroxide we created a new figure, Supplementary Figure 29. The figure illustrates that the same hydroxide generated when a water donates a hydrogen to the substrate is located at a suitable distance to also perform the deprotonation step during the IM₃ to IM₄ transition. We have also added the following text to describe this:

“The positioning of these strong hydrogen bonds allow for the necessary orientation of the hydroxide for both the IM₁–IM₂ and IM₂–IM₃ transitions (Supplementary Fig. 29).”

Is the following scenario feasible? — 6-Hydroxy group of 6OHFCoA is deprotonated by C2 and the hydrogen migrates to C2.

We thank the reviewer for proposing this scenario. The distance between C2 and the 6-hydroxy hydrogen of 6OHFCoA is in the range of 2.1–2.4 Å and thus, this scenario is geometrically feasible. To determine if the deprotonation of 6-hydroxy hydrogen by C2 is energetically feasible, we performed a constrained geometry scan with the distance between the 6-hydroxy hydrogen and C2 as the reaction coordinate. The reaction profile depicts two peaks of 12.8 kcal/mol corresponding to the breaking of the hydrogen bond between the 6-hydroxy hydrogen and His161, and a peak of 27.4 kcal/mol corresponding to the deprotonation. While the computed energy for the deprotonation is high making it less likely than our preferred pathway, we are not able to definitely rule out this possibility. We also think that this intramolecular deprotonation by C2 could explain why H161A mutant exhibits WT-level activity and that His161 is dispensable for the enzyme activity. To reflect the findings from this scenario, we have generated a Supplementary Figure 33 and added the following text to our manuscript:

“Moreover, we explored the dispensability of His161 as a catalytic base and calculated the reaction energetics for two alternative initial steps that utilize a solvent hydroxide as the base or an intramolecular hydrogen transfer of *o*-hydroxy hydrogen to C2 of 6OHFCoA... We also observe that the intramolecular deprotonation by the C2–C3 double bond is a possible pathway for His161-independent reaction (Supplementary Fig. 33).”

Additionally, I would like to see the energetic barrier to occurring spontaneous trans-cis isomerization and lactonization in water, if possible, at physiological pH for 6OHFCoA in plant.

We thank the reviewer for this comment. We reported the energetic cost of the spontaneous *trans-cis* isomerization in implicit solvent with dielectric constant of water in Supplementary Figure 22, which is a large energetic barrier of 37.9 kcal/mol. We noticed that we did not define “implicit solvent” in our text, so we changed our main text to clarify this:

“The constrained geometry scans of the substrates in implicit solvent with dielectric constant of water show that isomerization without C2 protonation leads to a large energetic cost of 37.9 kcal/mol in addition to the 6.6 kcal/mol energy of IM₁ (Supplementary Figs. 25b,c).”

To further demonstrate that C2 protonation is an essential preparative step to isomerization, we ran three replicate 250 ns molecular dynamics simulations of *Af*COSY with the 6OHFCoA substrate harboring either a sp³- or sp²-carbon at C2 (each substrate starting at a dihedral angle of 180° for the C1–C2–C3–C4

dihedral, representing the *s-trans* state). The sp^3 -substrate isomerized immediately to a dihedral of approximately 80° during the equilibration phase, whereas the sp^2 -substrate failed to isomerize. This result validates our experimental findings of the significance of C2 protonation in isomerization. We have included this result as Supplementary Fig. 26 and the following text in our manuscript:

“Moreover, in three replicates of 250 ns MD simulation of *Af*COSY with the sp^3 -substrate starting at a dihedral angle of 180° for the C1-C2-C3-C4 dihedral, which isomerized to a dihedral of approximately 80° immediately during the equilibration phase, whereas the sp^2 -substrate failed to isomerize (Supplementary Fig. 26).”

We also noticed a typo mistake in Supplementary Fig. 25— the starting dihedral angle representative of the *s-trans* state of 6OHFCoA for the C1-C2-C3-C4 dihedral should be 180° instead of 0° . We have made this change in Supplementary Fig. 25.

In addition, we have calculated the energetic barrier for the lactonization in implicit solvent without the protein environment as suggested. The calculated energetic barrier for the lactonization is illustrated in Supplementary Fig. 27 and the following text has been added to our manuscript:

“Upon isomerization, the proposed IM_4 -to- IM_5 lactonization can occur subsequently as showcased by the downhill energetics in the constrained geometry scan of the C-O bond in implicit solvent (Supplementary Fig 27).”

Other comments:

1) Authors should address a reason why 6-OH-ferulic acid was monitored, in some experiments (e.g., Fig. 3, Fig. S10), by its fragment ion (e.g., m/z 165) instead of deprotonated molecule (e.g., m/z 209 $[M-H]^-$). In relation to this, it would be appreciated if the formula of ion ($[M-H]^-$, $[M+1-H]^-$, etc.) is given to each m/z value in the text to facilitate interpreting the methods and results. Additionally, the description “162.04623 m/z corresponding to the $[M-H+D]^-$ value of umbelliferone” in the legend to Fig.S17 would be incorrect. Since monoisotopic mass of umbelliferone is 162.0, m/z value of $[M-H+D]^-$ should be 163 ($=162-1+2$).

In page 28, “multiple reaction monitoring at m/z transitions corresponding to 6OHFA (166.1 m/z to 151.1 m/z) and scopoletin (192.1 m/z to 177.1 m/z) with...”: The m/z for the precursor ions would be 165 and 191 for 6OHFA and scopoletin, respectively, when analyzed by negative ionization mode.

We thank the reviewer for this comment. 6OHFA was monitored by its fragment ion due to the ionization source that rapidly fragmented the compound causing the major peak detected was 165 m/z instead of 209 m/z $[M-H]^-$. We have added the following text in the Methods, Liquid-chromatography and mass-spectrometry section to address this:

“As 6OHFA and (D2)-6OHFA are readily fragmented under our MS collection methods, we monitored the abundance of their major fragment ions 165.05556 m/z and 167.06804 m/z , respectively, under negative ionization mode.”

Additionally, we thank the reviewer for catching our typo in Supplementary Figure 19 caption. 162.04623 m/z corresponds to the $[M-2H+D]^-$ instead of $[M-H+D]^-$ accounting for the H/D exchange onto C2 of umbelliferone. We have made this change in the figure caption and Supplementary Figure 19 to reflect the correction of this mistake.

We also acknowledge the error about multiple reaction monitoring precursor ions. The m/z transition for 6OHFA is indeed 165.1 m/z to 150.1 m/z and for scopoletin is 191.1 m/z to 176.1 m/z under negative ionization mode. We have changed this in the Methods, Liquid-chromatography and mass-spectrometry section.

2) In Fig. S23 and S24, it would be helpful if residue labels (one letter code and residue number) are shown for the representative residues, at least, those forming hydrogen bond with the ligand.

We thank the reviewer for this suggestion. We have updated the figures with labels for the depicted residues. As the Supplementary Figure numbers have changed, these are now reflected in Supplementary Figures 28 and 30.

3) *In legend to Fig. 3C, the definition of H/D exchange rate is somewhat confusing, because it is unclear what "total products" refers to. In the second and fourth panels of Fig. 3d, plural signals other than those originated from M+0, M+1, and M+2 appear to be present in some intensity. Are these ions also included in the products?*

We thank the reviewer for this question. The total products in Figure 3c refers to the total ion intensity for *m/z* values corresponding to (D2)-Scopoletin; **4** and (D1)-Scopoletin; **5**. The ion intensities of the plural signals in Figure 3d are not included in the calculations for H/D-exchange rate. To clarify this, we have made the following changes in Figure 3 caption:

"H/D exchange rate is defined as the percentage of **5** out of the total ion intensity for *m/z* values corresponding to **4** and **5**."

4) *Please describe the definition of Log₂(FC) in Fig. 2e.*

The following change has been made in the Figure 2e caption:

"Log₂ fold-change between the LC-HRAM-MS peak areas of coumarin ([M-H]⁻ = 191.03443 *m/z*) in 4CL1+AtCOSY (WT or mutant) samples and that of 4CL1-only samples are shown."

5) *In Fig. 4a and Fig. S26, the drawing of curved arrows showing electron flow is insufficient or incorrect. The arrow at the phenoxide moiety of IM4 (Fig. 4a) should be deleted. In contrast, Fig. S26 depicts only some of the necessary arrows.*

We thank the reviewer for catching these mistakes. We have made the corrections for Fig. 4a and Supplementary Fig. 26 (now Supplementary Figure 34).

6) *Accession No. of SbCOSY should be described. Although it may be appeared somewhere in the manuscript, I could not find it. In Fig. S28, it would be better to label the branch tips corresponding to AtCOSY, GmCOSY, StCOSY, and SbCOSY.*

We thank the reviewer for these suggestions. We updated Supplementary Figure 28 (now Supplementary Figure 36) with the labels for the branch tips corresponding to AtCOSY, GmCOSY, StCOSY, and SbCOSY. We also have described the Accession # of SbCOSY in Supplementary Figure 38 caption.

"... SbCOSY (NCBI Accession # XM_002437812.2)"

7) *In Abstract, "a unique proton exchange mechanism at the β-carbon" should be changed to "a unique proton exchange mechanism at the α-carbon".*

We thank the reviewer for catching this error. This change has been made in our abstract.

8) *The term "hydroxyl" for the functional group -OH is not recommended by IUPAC. "Hydroxy" should be used instead.*

The term "hydroxyl" have been changed to "hydroxy" used throughout our main text.

Reviewer #3 (Remarks to the Author):

Kim and coworkers describe in their manuscript that COSY - contrary to previous hypotheses - possesses an unconventional active-site configuration.

The manuscript is well-written and the results are very interesting for the scientific community. The conclusions of the authors are sound and based on *in-vitro* enzyme activity test with deuterium-labeled substrates, enzyme mutagenesis studies and *in silico* quantum mechanical cluster modeling. The newly discovered mechanism has a substantial lower activation energy for the *trans*-to-*cis* isomerization of the hydroxycinnamoyl-CoA substrates, (a critical rate-limiting step leading to coumarin production) as compared to the mechanism previously proposed in Vanholme 2019. The results discussed in the manuscript are in line with earlier observations of scopoletin biosynthesis with deuterated substrates as described by Bayoumi et al., 2008. An report that preceded the discovery of the COSY enzymes. In addition, they made several interesting observations that are worth noticing. i) *In vitro* enzyme tests and mutagenesis studies showed that the observed Ca²⁺ ion and oxidation of Cys308 to sulfenic acid are not acceleration enzyme activity and are therefore likely artifacts. ii) The Phe40 seems to be involved in substrate/product binding in AtCOSY, but the Phe34 in AtHCT (corresponding to Phe40 in AtCOSY) is a surface-exposed residue and thus not most likely involved in substrate/product binding. iii) Mutation analysis showed that His161, although conserved and thought to be involved in 6-OH deprotonation, seems to be dispensable for the COSY activity, suggesting that deprotonation is not rate-limiting in the full catalytic cycle of COSY. This is further supported by QM calculations. iv) The authors correctly point out that the function of COSY orthologs in monocots will be an interesting topic for future research.

I have one major comment, that is likely easily address by the authors:

It is unclear to me whether X-ray diffraction was used to determine the structure of COSY, or whether the structure was computationally optimized based on the HCT. Although the authors mention the name "X-ray structure" in the materials and methods section and acknowledgments, the structure seems to generated via computational tools. "The AtCOSY-apo structure was determined first by molecular replacement using the native HCT structure from *C. canephora* (PDB:4G0B) as the search model via the EMBL-HH Auto-Rickshaw. [...]" Next there is also a section on the crystallisation; "Crystals for AtCOSY-apo, H161Q-apo, scopoletin-bound, umbelliferone-bound, and CoA-bound were grown at 25 °C by hanging-drop vapor diffusion method. [...]"

In the results section the authors write that crystallization was only successful for the CoA bound COSY: "Despite multiple attempts of soaking and co-crystallization of AtCOSY with various substrate mimetics, including *p*-coumaroyl-CoA, feruloyl-CoA, dihydro-*p*-coumaroyl-CoA and cinnamoyl-CoA, we did not yield substrate-bound crystal structures. Instead, we were able to obtain the structure of AtCOSY in complex with free CoA, [...]"

In case the CoA-bound structure of COSY was determined via X ray, then the information about the machines and techniques used need to be mentioned in the materials and methods section. The failed(?) attempt to generate crystals of other crystals could be useful to be mentioned. In case no X rays were used, it must be made clear from the results section that the structure of COSY was determined based on the structure of *C. canephora* HCT.

We apologize for any confusion regarding the source of the COSY structure reported in our study. We used X-ray crystallography to obtain the AtCOSY-apo (PDB: 8DQO), AtCOSY-scopoletin (PDB: 8DQP), AtCOSY-umbelliferone (PDB: 8DQQ), and AtCOSY-CoA (PDB: 8DQR) structures that are deposited on RCSB PDB, which will be publicly available upon publication of this study. The apo structure was initially determined by molecular replacement using *C. canephora* HCT structure as a related search model. We also have reported unsuccessful cases for obtaining substrate-bound crystal structures in our main text:

"Despite multiple attempts of soaking and co-crystallization of AtCOSY with various substrate mimetics, including *p*-coumaroyl-CoA, feruloyl-CoA, dihydro-*p*-coumaroyl-CoA and cinnamoyl-CoA, we did not yield substrate-bound crystal structures."

To clarify this text, we have made the following changes in our Methods section under Crystallization and X-ray structure determination:

“All crystallographic diffraction data sets were collected at beamlines 24-ID-C and 24-ID-E of the Advanced Photon Source at the Argonne National Laboratory by single-wavelength anomalous diffraction methods.”

Minor comments:

1) A typo in the abstract: “courmarin production” “coumarin production”

Thank you for catching this error. We have made the correction in the main text.

2) 6OHFCoA is not defined

Thank you for catching this error. We have made the correction in the main text.

3) The concentrations of Zn²⁺, Ni²⁺, Mn²⁺, Co²⁺, Fe²⁺ or Cu²⁺ that are used for enzyme testing and their original chemical nature are not given.

Thank you for catching this error. We have made the correction in the caption of Supplementary Fig. 10:

“The enzyme assay was carried out by adding each of the following sources of divalent cations CaCl₂, ZnCl₂, NiCl₂, MnCl₂, CoCl₂, FeSO₄ and CuCl₂ at 100 μM each.”

4) I do not fully understand the colors used in the boxes of Figure 3b and d. Why do these seem to correspond with the color of the peaks shown in the chromatograms and spectra? Please add the explanation in the legend of that figure.

We thank the reviewer for pointing this out. We recognize that the boxes in Figures 3b and 3d are confusing. We deleted the boxes and replaced them with text corresponding to the enzymes used in the *in vitro* assays for each XIC trace. To further clarify the colors of the peaks shown in the chromatograms and spectra, we've added the following explanation in the Figure 3 caption:

“The colored circles represent the relevant substrate or product peaks in each XIC trace corresponding to the compound number in a.”

5) “Evaluation of the QM-optimized geometries for each intermediate revealed that the *s-cis* isomer is the highest-energy intermediate with an energy of 27.6 kcal/mol relative to the reactant state (R) (Supplementary Table 3).” The name *s-cis* isomer is not used in the figures and tables, instead IM3 is used. Please define that *s-cis* is the same as IM3 at this position in the manuscript. (I noted later that it is defined in the next paragraph.)

We appreciate the suggestion. The main text has been updated to connect the intermediate IM₃ to the *s-cis* conformational state.

“Evaluation of the QM-optimized geometries for each intermediate revealed the *s-cis* isomer (IM₃) to be the highest-energy intermediate with an energy of 27.6 kcal/mol relative to the reactant state (R) (Supplementary Table 3).”

REVIEWER COMMENTS

Reviewer #1 (Remarks to the Author):

Kim et al. uncovered the proton exchange-based isomerization and lactonization mechanism for a known coumarin synthase COSY. The work reported apo and complex structures with the products for COSY, and studied the mechanism using deuterium labeling, site-directed mutagenesis, and quantum mechanical cluster modeling. The mechanism was unique compared to known trans-cis isomerases to date. Overall most part of the manuscript is clearly written. However, the work lacks of critical data to support the unique mechanism in its current form.

1. The crystal structure in a complex with the genuine substrate is preferred to explain the proton exchange-based mechanism. Results from docking are not always reliable to predict the substrate binding, especially when it is vital to the mechanism. This also applies to explaining the functions of the key amino acids in the mutants.
2. The mechanism proposed is not fully supported by the mutagenesis, as H161 was nonessential and W371 variants did not abolish the activities. Clearly the reaction relies on other key amino acids which has not been revealed.
3. While the conversion happens spontaneously and is catalyzed by light, it is not clear which of the enzyme activity assays were carried out in dark. It is thus difficult to understand certain results especially in the H/D exchange experiments.
4. The difference between COSY and conventional BAHD-family enzymes is attractive for the broad readership of Nature Communications. Conclusions of active-site loop and key amino acids need support from the reconstructed enzymes, and the diversity of the substrate is required.

Minor points

- 1) The structures of (D1)-scopoletin and (D2)-scopoletin should be confirmed by NMR.
- 2) The authors should explain why 66% D₂O and 50% D₂O were used in different H/D exchange experiments.

Reviewer #2 (Remarks to the Author):

In this article, the authors investigated the catalytic mechanism of a recently identified unique acyltransferase, COSY, through crystallographic analyses, site-directed mutagenesis, enzyme assay using deuterium labelled substrates, and computational chemistry. This class of acyltransferase is the only BAHD enzyme known to date that catalyzes intramolecular acyl group transfer. Therefore, explicating the catalytic machinery would substantially contribute to the elucidation of the potential roles of BAHD enzymes in plant secondary metabolism and could lead to unveiling undiscovered biosynthetic pathways.

However, there are a few questions I would like you to answer.

According to the proposed catalytic mechanism (Fig. 4), water (or hydroxide ion) seems to be one of the key factors. Since I cannot access to the structural data presented in this article now, it is impossible to confirm whether a water molecule was observed at the appropriate position in the crystal. If there is not, do you have any other experimental data that imply the presence of a water molecule at the position?

In the scheme, a water molecule is deprotonated to generate a hydroxide anion during IM₂–IM₃ transition. Then, one of the C₂ hydrogen is deprotonated stereo selectively by a hydroxide ion which is stabilized by W371. The latter hydroxide would be identical to the former hydroxide. But I wonder if the hydroxide ion generated from water is located at the distance and angle suitable for the following deprotonation from C₂ in the rotated form of the reactant (IM₃). It may be helpful if distance between the water molecule and C₂-hydrogen/carbon is indicated in the corresponding figures. Of course, it

would be better to prepare a new figure focusing on this point.

Is the following scenario feasible? — 6-Hydroxy group of 6OHFCoA is deprotonated by C2 and the hydrogen migrates to C2.

Additionally, I would like to see the energetic barrier to occurring spontaneous trans-cis isomerization and lactonization in water, if possible, at physiological pH for 6OHFCoA in plant.

Other comments:

1) Authors should address a reason why 6-OH-ferulic acid was monitored, in some experiments (e.g., Fig. 3, Fig. S10), by its fragment ion (e.g., m/z 165) instead of deprotonated molecule (e.g., m/z 209 $[M-H]^-$). In relation to this, it would be appreciated if the formula of ion ($[M-H]^-$, $[M+1-H]^-$, etc.) is given to each m/z value in the text to facilitate interpreting the methods and results. Additionally, the description “162.04623 m/z corresponding to the $[M-H+D]^-$ value of umbelliferone” in the legend to Fig.S17 would be incorrect. Since monoisotopic mass of umbelliferone is 162.0, m/z value of $[M-H+D]^-$ should be 163 (=162-1+2).

In page 28, “multiple reaction monitoring at m/z transitions corresponding to 6OHFA (166.1 m/z to 151.1 m/z) and scopoletin (192.1 m/z to 177.1 m/z) with...”: The m/z for the precursor ions would be 165 and 191 for 6OHFA and scopoletin, respectively, when analyzed by negative ionization mode.

2) In Fig. S23 and S24, it would be helpful if residue labels (one letter code and residue number) are shown for the representative residues, at least, those forming hydrogen bond with the ligand.

3) In legend to Fig. 3C, the definition of H/D exchange rate is somewhat confusing, because it is unclear what “total products” refers to. In the second and fourth panels of Fig. 3d, plural signals other than those originated from M+0, M+1, and M+2 appear to be present in some intensity. Are these ions also included in the products?

4) Please describe the definition of $\text{Log}_{10}(\text{FC})$ in Fig. 2e.

5) In Fig. 4a and Fig. S26, the drawing of curved arrows showing electron flow is insufficient or incorrect. The arrow at the phenoxide moiety of IM4 (Fig. 4a) should be deleted. In contrast, Fig. S26 depicts only some of the necessary arrows.

6) Accession No. of SbCOSY should be described. Although it may be appeared somewhere in the manuscript, I could not find it.

In Fig. S28, it would be better to label the branch tips corresponding to AtCOSY, GmCOSY, StCOSY, and SbCOSY.

7) In Abstract, “a unique proton exchange mechanism at the β -carbon” should be changed to “a unique proton exchange mechanism at the α -carbon”.

8) The term “hydroxyl” for the functional group -OH is not recommended by IUPAC. “Hydroxy” should be used instead.

Reviewer #3 (Remarks to the Author):

Kim and coworkers describe in their manuscript that COSY - contrary to previous hypotheses - possesses an unconventional active-site configuration.

The manuscript is well-written and the results are very interesting for the scientific community. The conclusions of the authors are sound and based on in-vitro enzyme activity test with deuterium-labeled substrates, enzyme mutagenesis studies and in silico quantum mechanical cluster modeling. The newly discovered mechanism has a substantial lower activation energy for the trans-to-cis isomerization of the hydroxycinnamoyl-CoA substrates, (a critical rate-limiting step leading to coumarin production) as compared to the mechanism previously proposed in Vanholme 2019. The results discussed in the manuscript are in line with earlier observations of scopoletin biosynthesis with

deuterated substrates as described by Bayoumi et al., 2008. An report that preceded the discovery of the COSY enzymes.

In addition, they made several interesting observations that are worth noticing. i) In vitro enzyme tests and mutagenesis studies showed that the observed Ca^{2+} ion and oxidation of Cys308 to sulfenic acid are not acceleration enzyme activity and are therefore likely artifacts. ii) The Phe40 seems to be involved in substrate/product binding in AtCOSY, but the Phe34 in AtHCT (corresponding to Phe40 in AtCOSY) is a surface-exposed residue and thus not most likely involved in substrate/product binding. iii) Mutation analysis showed that His161, although conserved and thought to be involved in 6-OH deprotonation, seems to be dispensable for the COSY activity, suggesting that deprotonation is not rate-limiting in the full catalytic cycle of COSY. This is further supported by QM calculations. iv) The authors correctly point out that the function of COSY orthologs in monocots will be an interesting topic for future research.

I have one major comment, that is likely easily address by the authors:

It is unclear to me whether X-ray diffraction was used to determine the structure of COSY, or whether the structure was computationally optimized based on the HCT. Although the authors mention the name "X-ray structure" in the materials and methods section and acknowledgments, the structure seems to generated via computational tools. "The AtCOSY-apo structure was determined first by molecular replacement using the native HCT structure from *C. canephora* (PDB:4G0B) as the search model via the EMBL-HH Auto-Rickshaw. [...]" Next there is also a section on the crystallisation; "Crystals for AtCOSY-apo, H161Q-apo, scopoletin-bound, umbelliferone-bound, and CoA-bound were grown at 25 °C by hanging-drop vapor diffusion method. [...]"

In the results section the authors write that crystallization was only successful for the CoA bound COSY: "Despite multiple attempts of soaking and co-crystallization of AtCOSY with various substrate mimetics, including p-coumaroyl-CoA, feruloyl-CoA, dihydro-p-coumaroyl-CoA and cinnamoyl-CoA, we did not yield substrate-bound crystal structures. Instead, we were able to obtain the structure of AtCOSY in complex with free CoA, [...]"

In case the CoA-bound structure of COSY was determined via X ray, then the information about the machines and techniques used need to be mentioned in the materials and methods section. The failed(?) attempt to generate crystals of other crystals could be useful to be mentioned. In case no X rays were used, it must be made clear from the results section that the structure of COSY was determined based on the structure of *C. canephora* HCT.

Minor comments:

- 1) A typo in the abstract: "courmarin production" \diamond "coumarin production"
- 2) 6OHFCoA is not defined
- 3) The concentrations of Zn^{2+} , Ni^{2+} , Mn^{2+} , Co^{2+} , Fe^{2+} or Cu^{2+} that are used for enzyme testing and their original chemical nature are not given.
- 4) I do not fully understand the colors used in the boxes of Figure 3b and d. Why do theses seem to correspond with the color of the peaks shown in the chromatograms and spectra? Please add the explanation in the legend of that figure.
- 5) "Evaluation of the QM-optimized geometries for each intermediate revealed that the s-cis isomer is the highest-energy intermediate with an energy of 27.6 kcal/mol relative to the reactant state (R) (Supplementary Table 3)." The name s-cis isomer is not used in the figures and tables, instead IM3 is used. Please define that s-cis is the same as IM3 at this position in the manuscript. (I noted later that it is defined in the next paragraph.)

REVIEWER COMMENTS

Reviewer #1 (Remarks to the Author):

The authors have addressed most of my questions and added new data regarding the key catalytic residues. I have no more concerns against publication of the work except for the lack of NMR data for (D1)-scopoletin and (D2)-scopoletin.

We thank the reviewer for their comments. As explained in the previous revision response, it is challenging to obtain enough (D1)-scopoletin and (D2)-scopoletin in our enzyme assays for NMR due to the limited amount of D2-6OHFA substrate. Thus, we report alternative experimental evidence to support the C2 H/D-exchange for the production of (D1)-scopoletin via the AtCOSY WT assay against 2OHpCCoA in 66% D₂O, the *in planta* H/D exchange experiment, and the consistency of results as observed in a previous study, Bayoumi et al. (*Phytochemistry* vol. 69 2928-2936, 2008).

Reviewer #2 (Remarks to the Author):

The manuscript by Kim et al. has been revised carefully throughout the manuscript. My concerns and questions have been addressed properly. I believe the manuscript has been sufficiently improved to explain the mechanism by which the COSY can perform the novel function. The results presented here will be of great interest to the readership of Nature Communications.

We thank the reviewer for their recommendation to publish the manuscript on Nature Communications.

Reviewer #3 (Remarks to the Author):

As far as I can judge, the authors did a great job in addressing all comments of the reviewers, including mine. I have two comments on the newly added text. If these minor issues are sorted out, I recommend to accept the manuscript for publication.

1) I am confused with the new statement: "Among the active-site-lining residues examined by site-directed mutagenesis, we identified Leu374, when mutated to alanine, completely abolishes the proton exchange mechanism. [...] Moreover, the observation that the L374A mutant still retains some coumarin synthase activity implicates additional contributions of the COSY active site to accelerate coumarin production besides the proton exchange mechanism." In the first sentence, the authors claim that the proton exchange mechanism would be completely abolished in the L374A version. In the last sentence they claim that this mutant version still has activity. The claim in the last sentence is correct (as shown in Fig. 2e). So can the authors explain more clearly what they mean with the first sentence? A suggestion: "Among the active-site-lining residues examined by site-directed mutagenesis, we identified Leu374, when mutated to alanine, largely abolishes the COSY catalytic activity (Fig. 2e)."

We thank the reviewer for this comment and we apologize for any confusion regarding this statement. While AtCOSY-dependent coumarin synthase activity is primarily attributed to the proton exchange mechanism as postulated in Figure 4a, characterization of the L374A mutant revealed that other features of the AtCOSY active site also have a minor contribution to the overall catalytic activity of the enzyme. This conclusion was made based on our observation that L374A completely abolishes the H/D-exchange mechanism (Figure 3b,c.), but does not completely abolish AtCOSY-dependent scopoletin production (Figure 2e). This point was further discussed in the last two sentences of the paragraph.

We have revised the first sentence of the paragraph to clarify this confusion:

"Among the active-site-lining residues examined by site-directed mutagenesis, we identified Leu374, when mutated to alanine, completely abolishes the proton exchange mechanism (Fig. 3c), which is the main contributor to lowering the activation energy by the COSY active site."

2) On the question why not 100% D₂O was used, the authors answered that *A. thaliana* plants did not grow optimally in media prepared in 100% D₂O. However, in the material and methods they wrote “50% D₂O was used because *A. thaliana* plants did not grow on media prepared in 100% D₂O”. Can the authors please confirm that the plants did not grow at all, or otherwise change the text into “[...] *A. thaliana* plants did not grow optimally on media prepared in 100% D₂O”.

We thank the reviewer for catching this. Most *A. thaliana* seeds did not germinate on ½ MS media prepared in 100% D₂O, while some eventually germinated. Thus, we have changed the text to the following, and added a citation that reported similar observations:

“50% D₂O was used because *A. thaliana* plants did not germinate or grow optimally on media prepared in 100% D₂O, similar to previously reported⁶⁴.”

REVIEWERS' COMMENTS

Reviewer #3 (Remarks to the Author):

I would like to thank the authors for their further clarifications in the text. I recommend the editor to accept this great work for publication in Nature Communications.